# TRADE IN MINUTES! RATIONALITY-DRIVEN AGENTIC SYSTEM FOR QUANTITATIVE FINANCIAL TRADING

**Zifan Song**[1,2]  **Kaitao Song**[2]  **Guosheng Hu**[3]  **Ding Qi**[1]  **Junyao Gao**[1]  **Xiaohua Wang**[4]
**Dongsheng Li**[2]  **Cairong Zhao**[1*]

[1]Tongji University  [2]Microsoft Research Asia  [3]University of Bristol  [4]Fudan University

## ABSTRACT

Recent advancements in large language models (LLMs) and agentic systems have shown exceptional decision-making capabilities, revealing significant potential for autonomic finance. Current financial trading agents predominantly simulate anthropomorphic roles that inadvertently introduce emotional biases and rely on peripheral information, while being constrained by the necessity for continuous inference during deployment. In this paper, we pioneer the harmonization of strategic depth in agents with the mechanical rationality essential for quantitative trading. Consequently, we present *TiMi* (***Tra**de **i**n **Mi**nutes*), a rationality-driven multi-agent system that architecturally decouples strategy development from minute-level deployment. *TiMi* leverages specialized LLM capabilities of semantic analysis, code programming, and mathematical reasoning within a comprehensive *policy-optimization-deployment* chain. Specifically, we propose a two-tier analytical paradigm from macro patterns to micro customization, layered programming design for trading bot implementation, and closed-loop optimization driven by mathematical reflection. Extensive evaluations across 200+ trading pairs in stock and cryptocurrency markets empirically validate the efficacy of *TiMi* in stable profitability, action efficiency, and risk control under volatile market dynamics.

## 1 INTRODUCTION

Recent breakthroughs in large language models (LLMs) (OpenAI, 2023; Grattafiori et al., 2024) have demonstrated significant potential for solving complex tasks. Researchers are continuously advancing the fundamental capabilities of LLMs through innovations in model architectures (Cai et al., 2024; Guo et al., 2025), training paradigms (Ding et al., 2023; Wang et al., 2025), and data scaling (Song et al., 2024; Zhu et al., 2025). Concurrently, a systematic research paradigm is emerging: leveraging LLMs as core cognitive engines to construct agentic systems (Zhang et al., 2025; Hu et al., 2024) with autonomous decision-making and execution capabilities. This approach transcends the limitations of single-model improvements by integrating semantic understanding, logical reasoning, and tool utilization abilities into dynamic workflows through modular architectural design and strategic task decomposition, aiming to track long-term challenges in real-world scenarios.

This paper focuses on quantitative finance (Wilmott, 2013; Sun et al., 2023), where the required composite capabilities (*e.g.*, real-time decision-making, risk control, and strategy iteration) present a highly practical yet challenging domain for autonomous agent research. Classical rule-based strategies (Platen & Heath, 2006), while maintaining stable performance under specific market patterns, struggle to adapt to complex dynamics such as non-linear fluctuations and black swan events in the financial ecosystem. Notably, existing research on LLM-powered financial trading agents (Ding et al., 2024; Li et al., 2023; Xiao et al., 2025) emphasizes role-playing analysis and decision-making, including financial assistants and news-driven or debate-driven variants. Although these anthropomorphic approaches effectively leverage the strengths of LLMs in processing textual information, they pay less attention to the advancements in code programming and mathematical reasoning capabilities, which can be the key to achieving *mechanical rationality* in financial trading.

*"We don't let anyone predict the market—we let the models speak."* —— James Simons

---

*Corresponding author (zhaocairong@tongji.edu.cn). Work done during Zifan's internship at MSRA.

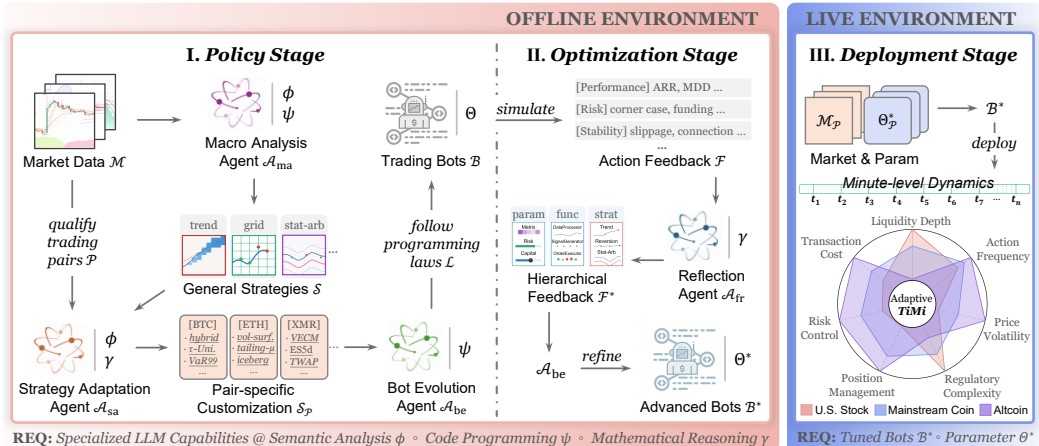

Figure 1: **Architecture of the proposed *TiMi* system comprising three stages: policy, optimization, and deployment.** *TiMi* implements a decoupling mechanism where the initial two stages develop and optimize prototype trading bots through offline simulations by leveraging specialized LLM capabilities, while the deployment stage executes thoroughly refined bots with tuned parameters in live trading. This paradigm separates complex reasoning from time-sensitive execution, enabling both *comprehensive strategy development* and *quantitative-level efficiency* across market dynamics.

To delve deeper, we identify three key aspects driving this exploration: (1) *market analysis paradigms* — the simulation of human trading organizations (*e.g.*, sentiment/news analysts, traders with varying risk preferences) in previous research inadvertently introduces interference from emotional biases and subjective judgments simulated by agents; (2) *supporting data selection* — unstructured peripheral information regarding target trading pairs (*e.g.*, heterogeneous news on social media, project reports) frequently contains misleading signals and temporal lags, which is particularly problematic for retail investors, as dependence on such publicly available information may lead to missed trading opportunities or substantial exposure to adverse market movements; (3) *system deployment efficiency* — the lengthy reasoning and negotiation among multiple agents significantly increase computational costs and action delays during practical deployment, which, in high-volatility trading environments, can manifest as execution slippage and opportunity costs.

In light of these considerations, we introduce *TiMi* (***T**rade **i**n **Mi**nutes*) depicted in Figure 1, a novel agentic system that achieves minute-level dynamic trading through rational decision-making. Regarding market analysis, we design top-level agents to capture and analyze patterns, deriving macro strategies from technical indicators, while specialized agents optimize strategies at the micro level based on specific trading pair characteristics. For data selection, we utilize objective technical indicators of target pairs (*e.g.*, volume and amplitude) with dynamically updated time windows to adapt to market fluctuations. To enhance deployment efficiency, we decouple analysis from execution by transforming strategies into programmatic trading bots through bot evolution agents (*i.e.*, Code LLMs). This approach enables minute-level quantitative trading with low latency, eliminating the computational costs and time consumption associated with continuous multi-agent inference. Essentially, we collect deployment feedback and employ reflection agents with reasoning capabilities, formulating mathematical problems (*e.g.*, linear programming) from representative cases to determine optimal parameters. These parameters are then submitted to bot evolution agents for hierarchical refinement across parameter, function, and strategy layers. Through this architecture, we effectively leverage the specialized capabilities of agents in semantic analysis, code programming, and mathematical reasoning, establishing a complete *closed-loop* system encompassing market analysis, strategy customization, programmatic deployment, and feedback iteration.

We perform live trading experiments on over 200 trading pairs across the U.S. stock index and cryptocurrency markets, reporting comprehensive metrics including Annual Rate of Return, Sharpe ratio, and Maximum Drawdown. The proposed *TiMi* demonstrates a competitive advantage among quantitative, ML/RL-based, and LLM-agent methods, particularly in challenging altcoin markets. Crucially, we present the systematic evolution of trading bots and conduct in-depth analytical studies with visualizing representative transactions from actual deployments, thereby examining the capacity of *TiMi* under various market dynamics. The core contributions of our work are threefold:

- We introduce *TiMi* (***T**rade **i**n **Mi**nutes*), a rationality-driven agentic system for quantitative financial trading that effectively leverages the complementary capabilities of different LLM variants across semantic analysis, code programming, and mathematical reasoning.

- Our *TiMi* system pioneers several key innovations: (1) strategic decoupling of strategy development from real-time deployment; (2) a two-tier analytical paradigm from macro patterns to micro customization; (3) a layered programming design for trading bot implementation; and (4) a closed-loop optimization system driven by mathematical reflection.

- Through comprehensive evaluations across 200+ diverse trading pairs, we empirically validate the efficacy of *TiMi* in profitability, deployment efficiency, and risk mitigation, offering an exploration for developing customizable agentic trading systems.

## 2 RATIONALITY-DRIVEN MULTI-AGENT SYSTEM

### 2.1 PRELIMINARY

We aim to develop an agentic system with a comprehensive *policy-deployment-optimization* chain to navigate market dynamics. Theoretically, each trading environment can be modeled as a tuple $(\mathcal{M}, \mathcal{W}, \mathcal{S}, \mathcal{F}, \mathcal{J})$, where $\mathcal{M}$ represents the market, $\mathcal{W}$ represents the targeted time window, $\mathcal{S}$ defines the strategy space, $\mathcal{F}$ denotes feedback signals, and $\mathcal{J}$ denotes evaluation functions. The trading system is expected to achieve: (1) analysis: $\mathcal{M} \times \mathcal{W} \to \mathcal{S}$, which transforms observed market patterns into trading strategies; (2) deployment: $\mathcal{M} \times \mathcal{S} \to \mathcal{F}$, which converts strategies into transactions and collects feedback during actions; and (3) optimization: $\mathcal{S} \times \mathcal{F} \to \mathcal{S}^*$, which refines strategies based on transaction feedback. Given a trading policy $\pi \in \mathcal{S}$ parameterized by $\Theta$, our *TiMi* is dedicated to maximizing $\mathcal{J}(\pi_\Theta)$ through a rationality-driven agentic system detailed in subsequent sections.

### 2.2 MULTI-AGENT ARCHITECTURE WITH DECOUPLED ANALYSIS AND DEPLOYMENT

Building upon the mechanical rationality, we formulate a multi-agent architecture leveraging specialized LLM capabilities in semantic analysis, code programming, and mathematical reasoning, thereby mitigating the inherent limitations of lacking adaptability (rule-based approaches) or introducing emotional biases (anthropomorphic simulations). Simultaneously, we advocate for decoupling analysis from deployment to separate strategy preparation from time-sensitive execution.

**Multi-agent design.** As presented on the left side of Figure 1, our *TiMi* comprises four specialized agents that interact in a coordinated workflow to transform market data into executable trading actions: (1) *macro analysis agent* $\mathcal{A}_{\mathrm{ma}}$ — identifies macro-level market patterns and formulates general trading strategies $\mathcal{S}$ based on technical indicators; (2) *strategy adaptation agent* $\mathcal{A}_{\mathrm{sa}}$ — customizes macro strategies $\mathcal{S}$ into pair-specific rules $\mathcal{S}_\mathcal{P}$ with initialized parameters $\Theta_\mathcal{P}$ by analyzing characteristics of trading pairs $\mathcal{P}$; (3) *bot evolution agent* $\mathcal{A}_{\mathrm{be}}$ — creates and optimizes programmatic trading bots $\mathcal{B}$ from trading strategies and feedback reflection; (4) *feedback reflection agent* $\mathcal{A}_{\mathrm{fr}}$ — reflects upon action feedback $\mathcal{F}$ to obtain more precise and hierarchical feedback $\mathcal{F}^*$ with refined parameters $\Theta^*$ *w.r.t.* $\mathcal{B}$. Let $\phi$, $\psi$, and $\gamma$ respectively represent the capabilities of semantic analysis, code programming, and mathematical reasoning, the complete *TiMi* system can be formulated as a composition of these agent functions:

$$\mathcal{A}_{\mathrm{ma}} \circ \phi \circ \psi : \mathcal{M} \times \mathcal{W} \to \mathcal{S}, \quad \mathcal{A}_{\mathrm{sa}} \circ \phi \circ \gamma : \mathcal{S} \times \mathcal{P} \to \mathcal{S}_\mathcal{P} \times \Theta_\mathcal{P},$$
$$\mathcal{A}_{\mathrm{be}} \circ \psi : \mathcal{S} \times \Theta \times \mathcal{L} \to \mathcal{B}, \quad \mathcal{A}_{\mathrm{fr}} \circ \gamma : \mathcal{B} \times \mathcal{F} \times \Theta \to \mathcal{F}^* \times \Theta^*, \quad (1)$$
$$\mathcal{T}(\mathcal{M}, \mathcal{W}) = \mathcal{A}_{\mathrm{be}}(\mathcal{A}_{\mathrm{sa}}(\mathcal{A}_{\mathrm{ma}}(\mathcal{M}, \mathcal{W}), \mathcal{P}), \mathcal{L})(\mathcal{M}; \mathcal{A}_{\mathrm{fr}}(\mathcal{B}, \mathcal{F}, \Theta)).$$

where $\circ$ indicates a conceptual combination of functionalities (*i.e.*, the agents perform defined mapping tasks by invoking embedded core capabilities), $\mathcal{T}(\mathcal{M}, \mathcal{W})$ represents the system operating on the market $\mathcal{M}$ with the time window $\mathcal{W}$, and $\mathcal{L}$ denotes programming laws (detailed in Section 2.4).

**Decoupling mechanism.** We achieve the decoupling of analysis and deployment through a three-stage process: (1) *policy stage* — complex reasoning and strategy development occur in an offline environment, fully leveraging the capabilities of specialized agents, including actions of $\mathcal{A}_{\mathrm{ma}}$, $\mathcal{A}_{\mathrm{sa}}$, and $\mathcal{A}_{\mathrm{be}}$ to generate prototype trading bots $\mathcal{B}$ with initialized parameter $\Theta$; (2) *optimization stage* — the prototype bots undergoes simulation in offline environments (*e.g.*, live or historical markets) to gather feedback $\mathcal{F}$ including technical execution traceback and risk corner cases, to iteratively conduct offline agent interactions and achieve advanced trading bots $\mathcal{B}^* = \mathcal{A}_{\mathrm{be}}(\mathcal{B}; \mathcal{A}_{\mathrm{fr}}(\mathcal{B}, \mathcal{F}, \Theta))$; (3)

*deployment stage* — following thorough optimization, the trading bot that has successfully passed simulation tests can be deployed in live trading environments with low latency and execution costs (a concrete implementation is discussed in Section 3). This mechanism eliminates the requirement for continuous model inference during actual deployment and creates an efficiency advantage, quantified as $\eta = \frac{c_{\text{agent}} \times n}{c_{\text{policy}} + c_{\text{optimization}} + c_{\text{bot}} \times n}$, where $c_{\text{agent}}/c_{\text{bot}}$ is the agent/bot inference cost per trade, $n$ is the number of trading actions, and $c_{\text{analysis}}$ is the offline analysis cost. As $n$ increases in high-volatility markets, the efficiency ratio approaches $\lim_{n \to \infty} \eta = \frac{c_{\text{agent}}}{c_{\text{bot}}}$. Given that typically $c_{\text{bot}} \ll c_{\text{agent}}$, this represents an *efficiency* and *responsiveness* improvement that scales with trading frequency. Concurrently, the decoupling enables in-depth strategy refinement during the optimization stage without temporal constraints, contributing to enhanced efficacy and more robust trading performance.

## 2.3 ANALYTICAL PARADIGM FROM MACRO PATTERNS TO MICRO CUSTOMIZATION

We implement a two-tier paradigm for strategy initialization: from market-wide analysis to pair-specific customization. It is designed to offer advantages in both *statistical significance* and *strategic adaptability* compared to monolithic approaches.

**Macro strategy analysis.** In theory (Hasbrouck, 2007; Lo et al., 2000), financial markets fundamentally possess periodic behavioral patterns under specific conditions (*e.g.*, within short-term time windows), which can be identified through a combination of technical indicators and statistical methods. Consequently, the foundation of our system is the macro analysis agent $\mathcal{A}_{\text{ma}}$, which performs rational analysis of market patterns. Initialized through the definition of technical indicators $\mathcal{I}$, $\mathcal{A}_{\text{ma}}$ captures the state space encompassing all observable market conditions across time scales $\mathcal{W}$. This enables the generation of a general strategy set $\mathcal{S}$, specifically oriented toward patterns demonstrating statistical significance. Formally, the operational mechanism of $\mathcal{A}_{\text{ma}}$ can be expressed as:

$$\mathcal{A}_{\text{ma}}(\mathcal{M}, \mathcal{W}; \mathcal{I}) = \phi(\{\psi_i(\mathcal{M}, w) | w \in \mathcal{W}, i \in \mathcal{I}\}) \to \mathcal{S}. \tag{2}$$

here, the function $\psi_i(\mathcal{M}, w)$ denotes a programming process that extracts relevant market data within time window $w$ and applies indicator $i$ to transform this data into analytically useful features.

**Pair-specific customization.** Different trading pairs often exhibit heterogeneous behaviors due to their unique characteristics. To track this, we introduce strategy adaptation agent $\mathcal{A}_{\text{sa}}$ that systematically refines the general strategy set for specific trading pairs. Our methodology employs a two-step process: initially, we perform semantic analysis $\phi(\mathcal{S}, p) | p \in \mathcal{P} \to \mathcal{S}_p$ to select and adapt strategies from the general set $\mathcal{S}$, thereby creating pair-specific strategy candidates $\mathcal{S}_p$; subsequently, we conduct mathematical reasoning $\gamma(\mathcal{S}_p, p) | p \in \mathcal{P} \to \Theta_p$ to optimize the parameter set $\Theta_p$ for these strategies. Crucially, this customization encompasses strategy prioritization based on historical performance, parameter calibration tailored to pair-specific volatility profiles, and adaptive risk management rules that account for critical factors such as market liquidity.

## 2.4 IMPLEMENTATION OF TRADING BOTS WITH LAYERED PROGRAMMING DESIGN

To transform strategic insights into executable trading bots, we implement a layered programming policy that enhances modularity and facilitates systematic refinement. Our bot evolution agent $\mathcal{A}_{\text{be}}$ constructs trading bots $\mathcal{B}$ by decomposing them into three hierarchical layers: strategy, function, and parameter. The strategy layer encapsulates decision-making logic derived from $\mathcal{S}_p$, including signal generation, position sizing, and entry/exit criteria. The functional layer provides computational mechanisms required by the strategy, implementing technical indicators, data preprocessing, and order execution routines that are reusable across different strategies. The parameter layer manages the adjustable parameters that fine-tune the behavior of the trading strategy and its functions. This architecture enables $\mathcal{A}_{\text{be}}$ to efficiently transform pair-specific strategies into algorithmic procedures while facilitating the decoupling mechanism between policy and development stages.

**Programming laws.** We present three core laws $\mathcal{L}$ that govern the code programming $\psi$ of $\mathcal{A}_{\text{be}}$: (1) *functional cohesion law* — each functional component must address exactly one responsibility; (2) *unidirectional dependency law* — dependencies flow strictly from higher to lower layers; and (3) *parameter externalization law* — all adjustable values must be extracted from implementation code and centrally managed. These principles are designed for $\mathcal{A}_{\text{be}}$ to enable systematic construction of trading bots that support the feedback-driven refinement process initiated by $\mathcal{A}_{\text{fr}}$ while maintaining architectural integrity across optimization cycles.

## 2.5 CLOSED-LOOP OPTIMIZATION DRIVEN BY MATHEMATICAL REFLECTION

In the optimization stage, trading bots are simulated in live or historical markets to periodically collect action feedback $\mathcal{F}$, encompassing trading performance metrics, risk event records, and execution statistics. The feedback reflection agent $\mathcal{A}_{\mathrm{fr}}$ deconstructs this feedback and formulates precise optimization plans, which are then transmitted to $\mathcal{A}_{\mathrm{be}}$ for programmatic refinement. In this way, we establish a rationality-driven evolutionary process toward a robust and reliable system.

**Mathematical reasoning for parameter solving.** Our feedback reflection agent $\mathcal{A}_{\mathrm{fr}}$ employs mathematical reasoning $\gamma$ in a three-step optimization process: first organizing risk scenarios from feedback $\mathcal{F}$ and transforming them into linear programming problems; then solving for the feasible parameter solution space; and finally optimizing parameters within the constrained space to maximize performance. This optimization can be formally expressed as (exemplified in Appendix A):

$$\Theta^* = \arg\max_{\Theta \in \mathcal{C}(\Theta)} \sum \omega_i \mathcal{J}_i(\Theta, \mathcal{F}) \text{ s.t.} \quad \mathcal{C}(\Theta) = \{\Theta \in \mathbb{R}^n \mid \mathbf{A}(\mathcal{R})\Theta \preceq \mathbf{b}(\mathcal{R})\}. \tag{3}$$

where $\mathcal{C}(\Theta)$ defines the feasible parameter space, $\omega_i$ and $\mathcal{J}_i$ denote $i$-th objective weight and evaluation metric (*e.g.*, win rate) respectively, while $\mathbf{A}(\mathcal{R})$ and $\mathbf{b}(\mathcal{R})$ represent the constraint matrix and threshold vector derived from risk scenarios $\mathcal{R} = \gamma(\mathcal{F})$, implementing parameter restrictions through component-wise inequality $\preceq$. Critical to this process is the ability of $\mathcal{A}_{\mathrm{fr}}$ to recognize trade-offs between competing objectives and establish Pareto-efficient parameter configurations.

**Hierarchical optimization.** We propose a hierarchical optimization scheme that propagates refinements from the parameter level (*i.e.*, the parameter layer of $\mathcal{B}$) upward through the trading system. At the parameter level, we focus on fine-tuning numerical values within constraints. When parameter adjustments prove insufficient to meet requirements (*e.g.*, failing risk simulations), $\mathcal{A}_{\mathrm{fr}}$ escalates to function level and substitutes algorithmic components. The highest level of intervention occurs at the strategy layer, where fundamental decision-making rules encoded in $\mathcal{S}_p$ undergo structural modifications. This tiered manner, as exemplified in Figure 2, offers dual advantages: it adheres to the principle of *minimal intervention* by prioritizing lower-level adjustments that preserve strategic continuity, and it establishes a natural complexity progression that enables testing less disruptive modifications before implementing more fundamental changes.

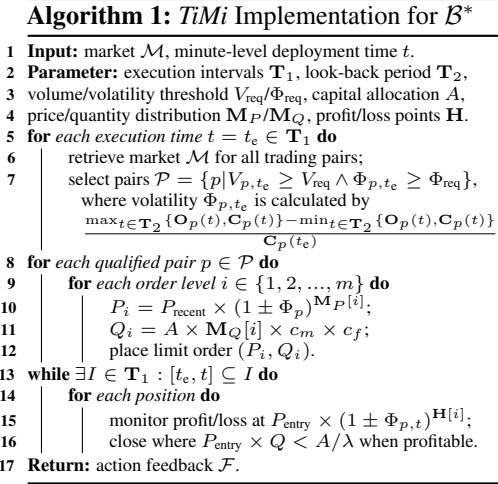

**Algorithm 1:** *TiMi* Implementation for $\mathcal{B}^*$

1 **Input:** market $\mathcal{M}$, minute-level deployment time $t$.
2 **Parameter:** execution intervals $\mathbf{T}_1$, look-back period $\mathbf{T}_2$,
3 volume/volatility threshold $V_{\mathrm{req}}/\Phi_{\mathrm{req}}$, capital allocation $A$,
4 price/quantity distribution $\mathbf{M}_P/\mathbf{M}_Q$, profit/loss points $\mathbf{H}$.
5 **for** *each execution time* $t = t_\mathrm{e} \in \mathbf{T}_1$ **do**
6     retrieve market $\mathcal{M}$ for all trading pairs;
7     select pairs $\mathcal{P} = \{p | V_{p,t_\mathrm{e}} \geq V_{\mathrm{req}} \wedge \Phi_{p,t_\mathrm{e}} \geq \Phi_{\mathrm{req}}\}$,
    where volatility $\Phi_{p,t_\mathrm{e}}$ is calculated by
    $\frac{\max_{t \in \mathbf{T}_2}\{\mathbf{O}_p(t), \mathbf{C}_p(t)\} - \min_{t \in \mathbf{T}_2}\{\mathbf{O}_p(t), \mathbf{C}_p(t)\}}{\mathbf{C}_p(t_\mathrm{e})}$.
8 **for** *each qualified pair* $p \in \mathcal{P}$ **do**
9     **for** *each order level* $i \in \{1, 2, ..., m\}$ **do**
10         $P_i = P_{\mathrm{recent}} \times (1 \pm \Phi_p)^{\mathbf{M}_P[i]}$;
11         $Q_i = A \times \mathbf{M}_Q[i] \times c_m \times c_f$;
12         place limit order $(P_i, Q_i)$.
13 **while** $\exists I \in \mathbf{T}_1 : [t_\mathrm{e}, t] \subseteq I$ **do**
14     **for** *each position* **do**
15         monitor profit/loss at $P_{\mathrm{entry}} \times (1 \pm \Phi_{p,t})^{\mathbf{H}[i]}$;
16         close where $P_{\mathrm{entry}} \times Q < A/\lambda$ when profitable.
17 **Return:** action feedback $\mathcal{F}$.

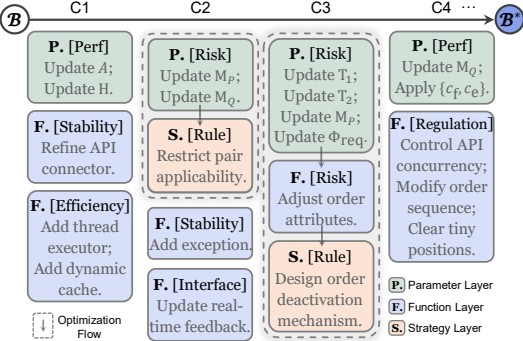

Figure 2: **Evolution map of the trading bots $\mathcal{B}$.** We present deliberate optimization cycles (C1-C4) *w.r.t.* parameter, function, and strategy layers — showcasing how hierarchical optimization *progressively* drives sophisticated trading capabilities.

## 3 TRADE IN MINUTES

In this section, we present a concrete implementation of advanced trading bots $\mathcal{B}^*$ (detailed in Algorithm 1) that demonstrates the practical deployment of the proposed *TiMi* system, with particular emphasis on parameter configuration, order execution logic, position management, and risk control.

**Parameter configuration.** *TiMi* establishes crucial parametric variables governing trading operations, including temporal constraints $\mathbf{T}_1, \mathbf{T}_2$ defining minute-level execution intervals and volatility look-back period respectively, and risk allocation amount $A$ for capital distribution. Additionally, minimum

trading volume threshold $V_{\text{req}}$ ensures sufficient liquidity, while parameter $\Phi_{\text{req}}$ serves as the volatility criterion for trading pair qualification. The system incorporates matrix-based parameters $\mathbf{M}_P = [p_1, p_2, ..., p_m]$ and $\mathbf{M}_Q = [q_1, q_2, ..., q_m]$ controlling order distribution alongside quantity scaling coefficient $\{c_{\text{m}}, c_{\text{f}}, c_{\text{e}}\}$ for adjustment of market capitalization, funding rates, and position entry. Here, profit/loss thresholds $\mathbf{H} = [h_1, h_2, ..., h_k]$ are adopted for position management.

**Order execution logic.** Starting with retrieving market data through API endpoints, the system calculates essential indicators within the execution period $[t_{\text{e}}, t_{\text{e}} + \Delta t_1] \in \mathbf{T}_1$, including price metrics, volatility indices, and funding rates. Then trading pairs $\mathcal{P} = \{p | V_{p,t_{\text{e}}} \geq V_{\text{req}} \land \Phi_{p,t_{\text{e}}} \geq \Phi_{\text{req}}\}$ satisfying volume and volatility requirements are qualified by filtering rules, estimated through $\Phi_{p,t_{\text{e}}} = \frac{\max_{t \in \mathbf{T}_2}\{\mathbf{O}_p(t), \mathbf{C}_p(t)\} - \min_{t \in \mathbf{T}_2}\{\mathbf{O}_p(t), \mathbf{C}_p(t)\}}{\mathbf{C}_p(t_{\text{e}})}$, where $\mathbf{T}_2 = \{t_{\text{e}} - \tau, t_{\text{e}} - \tau + \Delta t_2, ..., t_{\text{e}}\}$ represents a sequence of time points determined by the estimation window $\tau$ and time step $\Delta t_2$, and $\mathbf{O}_p(t)/\mathbf{C}_p(t)$ represents the opening/closing price of the K-line corresponding to pair $p$ during interval $[t - \Delta t_2, t]$. Specifically, *TiMi* implements a precision-engineered grid strategy with minute-level dynamics, placing orders at optimized price levels $P_i = P_{\text{recent}} \times (1 \pm \Phi)^{\mathbf{M}_P[i]}$ for selected pairs. Order quantity $Q_i$ is calculated by $Q_i = A \times \mathbf{M}_Q[i] \times c_{\text{m}} \times c_{\text{f}}$. For positioned assets, *TiMi* applies a proportional scaling factor $(\frac{P}{P_{\text{entry}}})^{c_{\text{e}}}$ to dynamically adjust allocation when positions move.

**Position management.** During deployment, the system continuously monitors positions and market dynamics. Upon reaching profit/loss thresholds $P_{\text{entry}} \times (1 \pm \Phi)^{\mathbf{H}[i]}$, *TiMi* executes partial position closures through a progressive realization manner. Meanwhile, positions where $P_{\text{entry}} \times Q < A/\lambda$ (with $\lambda$ as position size divisor) are automatically closed when profitable, optimizing capital efficiency.

**Risk control.** *TiMi* integrates sophisticated mechanisms for risk mitigation, ensuring robust trading across varied market dynamics. Essentially, the system employs mathematically optimized parameter matrices $\mathbf{M}_P$ and $\mathbf{M}_Q$ that have undergone rigorous refinement through the feedback reflection agents $\mathcal{A}_{\text{fr}}$, with these parameters solved within a constrained feasible solution space derived from extensive risk scenario simulations. Simultaneously, capital allocation is precisely governed by the $A$ parameter, limiting exposure per asset and preventing concentration risk. Additionally, our *TiMi* performs price deviation control to prevent order placement during abnormal market conditions when significant discrepancies exist between the latest and mark prices.

## 4 EXPERIMENTS

### 4.1 IMPLEMENTATION DETAILS

**Backbone LLMs.** As articulated in Section 2, the core design philosophy of *TiMi* leverages specialized LLM capabilities. We strategically adopt DeepSeek-V3 for semantic analysis, Qwen2.5-Coder-32B-Instruct for code programming, and DeepSeek-R1 for mathematical reasoning. Besides, we develop a hybrid implementation that combines local inference (small models) and API-based inference (large models), facilitating flexible upgrading and optimal performance-efficiency trade-offs.

**Agentic Implementation.** We develop a hybrid communication protocol that combines XML-based message envelopes with JSON payloads to facilitate inter-agent data exchange. The XML layer encapsulates essential metadata (*e.g.*, sender identity and contextual domain), while the JSON payload carries domain-specific content. Our agents operate within a deterministic environment equipped with system-level capabilities, including local file operations and verifiable API invocations. Crucially, we implement procedural posterior checks where *TiMi* programmatically verifies generated scripts and mathematical solutions within controlled sandboxes, capturing execution tracebacks to ensure computational outputs and parameter derivations satisfy predefined constraints before deployment.

**Deployment.** Benefiting from our decoupling mechanism, *TiMi* requires a CPU-only runtime environment during the deployment stage. The trading bots developed by the agents $\mathcal{A}_{\text{be}}$ and $\mathcal{A}_{\text{fr}}$ are implemented in Python and integrated with exchange APIs through standardized connectors. Additionally, *TiMi* achieves error-handling routines to manage connectivity issues, rate limits, and unexpected market conditions, ensuring operational continuity under suboptimal circumstances.

**Simulation and live trading.** We conduct extensive experiments across both U.S. stock index futures and cryptocurrency markets to evaluate the versatility and robustness of *TiMi* under diverse market conditions. We implement a progressive validation: initial strategy development using historical data, followed by trading simulation with real-time market data, and culminating in live trading evaluation.

Table 1: ***Backtesting* comparison** on 2024 historical data across U.S. stock index futures and cryptocurrency markets. Optimal/suboptimal results are indicated by **bold**/underline, respectively.

| Method | U.S. Stock Index Futures | | | Mainstream Coin Futures | | | Altcoin Futures | | |
|---|---|---|---|---|---|---|---|---|---|
| | ARR%↑ | SR↑ | MDD%↓ | ARR%↑ | SR↑ | MDD%↓ | ARR%↑ | SR↑ | MDD%↓ |
| *Quantitative Methods* | | | | | | | | | |
| MACD | 3.8 | 0.42 | 14.5 | 12.6 | 0.84 | 18.4 | 4.2 | 0.36 | 32.8 |
| Momentum | 8.0 | 0.73 | 16.8 | 15.7 | 0.93 | 21.4 | -2.8 | -0.40 | 38.3 |
| Grid Trading | -2.8 | -0.35 | 10.2 | -8.3 | -0.72 | 22.6 | 9.4 | 0.72 | 18.7 |
| Pairs Trading | 2.3 | 0.40 | **7.4** | -6.8 | -0.67 | 16.3 | -5.7 | -0.46 | **14.5** |
| ETF&PCA | 4.6 | 0.49 | 13.9 | 7.2 | 0.62 | 15.8 | 6.3 | 0.69 | 17.2 |
| TSMOM | **10.5** | 0.81 | 19.7 | **18.4** | 1.15 | 17.2 | 1.5 | 0.05 | 41.6 |
| OFI | 2.1 | 0.34 | 11.8 | 8.9 | 0.66 | 19.3 | 7.8 | 0.64 | 23.9 |
| *ML&RL Methods* | | | | | | | | | |
| LSTM | 3.2 | 0.36 | 15.3 | 9.4 | 0.88 | 17.8 | -16.8 | -0.89 | 25.4 |
| DQN | 8.3 | 0.69 | 18.6 | 9.7 | 0.83 | 20.5 | -9.3 | -0.80 | 38.7 |
| DDPG | 5.4 | 0.52 | 17.4 | 14.8 | 1.09 | 14.0 | 8.6 | 0.65 | 26.2 |
| Autoformer | 4.9 | 0.43 | 16.5 | 13.2 | 0.97 | 16.4 | 17.5 | 0.98 | 24.8 |
| PatchTST | 6.7 | 0.58 | 18.1 | 12.5 | 0.90 | 17.6 | 11.2 | 0.82 | 23.5 |
| *LLM-based Agents* | | | | | | | | | |
| FinGPT | 5.8 | 0.57 | 15.7 | 6.6 | 0.59 | 19.8 | 7.4 | 0.61 | 31.2 |
| FinMem | 5.2 | 0.54 | 14.8 | 11.3 | 0.96 | 17.4 | -8.9 | -0.60 | 19.6 |
| TradingAgents | 6.3 | 0.60 | 16.2 | 17.9 | 1.12 | 20.3 | 9.7 | 0.72 | 29.6 |
| *TiMi (ours)* | 8.9 | **0.84** | 10.5 | 16.5 | **1.25** | 12.1 | 23.7 | **1.27** | 26.0 |

**Evaluation metrics.** The primary metrics include Annual Rate of Return (ARR), which measures the change in investment value over a year as $\mathrm{ARR} = \frac{V_{\text{final}} - V_{\text{initial}}}{V_{\text{initial}}}$, where $V_{\text{final}}$ and $V_{\text{initial}}$ represent final and initial values; Sharpe Ratio (SR), quantifying excess return per unit of risk as $\mathrm{SR} = \frac{\overline{R} - R_f}{\sigma_P}$, where $\overline{R}$ is average portfolio return, $R_f$ is risk-free rate, and $\sigma_P$ is standard deviation of excess return; and Maximum Drawdown (MDD), representing the largest peak-to-trough decline as $\mathrm{MDD} = \frac{V_{\text{trough}} - V_{\text{peak}}}{V_{\text{peak}}}$, where $V_{\text{peak}}$ and $V_{\text{trough}}$ are the highest value before and lowest value after the largest drop.

**Baselines.** We compare *TiMi* against three representative categories: (1) *quantitative methods*, including MACD (Wang & Kim, 2018) optimized by historical volatility, momentum strategy (Jegadeesh & Titman, 1993), grid trading (Griffin et al., 2003), pairs trading (Gatev et al., 2006), ETF&PCA-based statistical arbitrage (Avellaneda & Lee, 2010), time-series momentum (TSMOM) (Moskowitz et al., 2012), and order flow imbalance (OFI) strategy (Cont & De Larrard, 2013); (2) *ML&RL methods*, spanning time-series forecasting (LSTM (Sunny et al., 2020), Autoformer (Wu et al., 2021), PatchTST (Nie et al., 2022)) and reinforcement learning (DQN (Mnih et al., 2013), DDPG (Liu et al., 2020)); (3) *LLM-based agents*, including news-driven FinGPT (Liu et al., 2023), memory-augmented FinMem (Yu et al., 2024a), and multi-agent TradingAgents (Xiao et al., 2025).

## 4.2 EMPIRICAL RESULTS

**Backtesting comparison.** We evaluate on 2024 historical data to establish baseline performance. As detailed in Table 1, trend-following methods (*e.g.*, TSMOM) capitalize on ETF-driven trends in mainstream coin markets, while LLM-agents exploit news and potential posterior information. Crucially, *TiMi* harmonizes high profitability with rigorous risk control, achieving superior stability and risk-adjusted returns (notably SR 1.27 in Altcoins). This highlights the robustness of our system in high-volatility, reflexive assets where traditional momentum or pure semantic analysis struggle.

**Live trading comparison.** Table 2 and Table 3 present comprehensive performance metrics evaluated in live trading environments. *TiMi* appears to outperform competing approaches, achieving ARR

Table 2: **Data (type&duration) requirement** and **Sortino Ratio comparison (altcoin)**. M: market indicators; N: peripheral news.

| Method | Data Req. | Sortino↑ |
|---|---|---|
| Grid | M > 30m | 0.16 |
| ETF&PCA | M > 7d | -0.33 |
| DDPG | M > 12h | 0.57 |
| PatchTST | M > 3d | 0.67 |
| FinMem | M&N > 1d | 0.41 |
| TradingAgents | M&N > 3d | 0.58 |
| *TiMi (ours)* | M > 4h | **0.91** |

Table 3: *Live trading* **comparison of mainstream methods** across U.S. stock index futures and cryptocurrency markets from 2025 January to April. TF and $N_{\mathcal{P}}$ represent the trading frequency and number of supported trading pairs for each method ("$*$" denotes estimated results from partial experiments). The optimal and suboptimal results are indicated by **bold** and underline, respectively.

| Method | U.S. Stock Index Futures | | | Mainstream Coin Futures | | | Altcoin Futures | | | $N_{\mathcal{P}}\uparrow$ | TF |
|---|---|---|---|---|---|---|---|---|---|---|---|
| | ARR%↑ | SR↑ | MDD%↓ | ARR%↑ | SR↑ | MDD%↓ | ARR%↑ | SR↑ | MDD%↓ | | |
| *Quantitative Methods* | | | | | | | | | | | |
| MACD | 2.1 | 0.32 | 22.4 | -5.9 | -0.66 | 38.3 | -12.5 | -0.85 | 41.3 | **213** | *daily* |
| Momentum | 1.5 | 0.23 | 25.5 | -6.2 | -0.58 | 31.0 | -8.4 | -0.67 | 37.5 | **213** | *daily* |
| Grid Trading | 3.2 | 0.42 | 17.2 | 3.2 | 0.25 | 25.9 | 1.8 | 0.15 | 28.4 | **213** | *hourly* |
| Pairs Trading | 0.8 | 0.08 | **11.0** | 2.8 | 0.22 | 27.4 | 4.5 | 0.49 | 25.6 | **213** | *daily* |
| ETF&PCA | 4.1 | 0.50 | 19.1 | -2.5 | -0.26 | **22.3** | -4.8 | -0.31 | 27.3 | 75 | *minute* |
| TSMOM | 3.8 | 0.44 | 24.9 | -9.5 | -0.77 | 40.8 | -10.2 | -0.78 | 42.9 | **213** | *daily* |
| OFI | -1.9 | -0.18 | 18.4 | 5.2 | 0.58 | 27.8 | 5.4 | 0.52 | 29.3 | **213** | *second* |
| *ML&RL Methods* | | | | | | | | | | | |
| LSTM | 1.2 | 0.12 | 18.4 | 1.8 | 0.14 | 28.5 | 2.8 | 0.26 | 28.2 | 70* | *daily* |
| DQN | 1.7 | 0.11 | 25.2 | -1.0 | -0.06 | 31.7 | -2.3 | -0.18 | 39.0 | 70* | *daily* |
| DDPG | 5.1 | 0.53 | 22.7 | 5.8 | 0.63 | 27.9 | 5.9 | 0.54 | 38.1 | 150* | *daily* |
| Autoformer | 4.4 | 0.48 | 21.1 | 4.9 | 0.47 | 28.4 | 8.3 | 0.66 | 42.5 | 120* | *daily* |
| PatchTST | 5.5 | 0.62 | 22.8 | 2.7 | 0.25 | 29.0 | 6.4 | 0.63 | 35.4 | 120* | *daily* |
| *LLM-based Agents* | | | | | | | | | | | |
| FinGPT | 5.1 | 0.57 | 22.6 | -3.7 | -0.31 | 29.5 | -6.2 | -0.60 | 30.6 | 81 | *daily* |
| FinMem | 3.6 | 0.45 | 19.7 | 4.4 | 0.45 | 27.3 | 3.8 | 0.39 | **23.7** | 50* | *daily* |
| TradingAgents | 4.8 | 0.50 | 20.4 | 5.4 | 0.63 | 25.6 | 5.5 | 0.57 | 28.3 | 28* | *daily* |
| *TiMi (ours)* | **6.4** | **0.74** | 20.3 | **8.0** | **0.79** | 25.1 | **13.7** | **0.86** | 32.8 | **213** | *minute* |

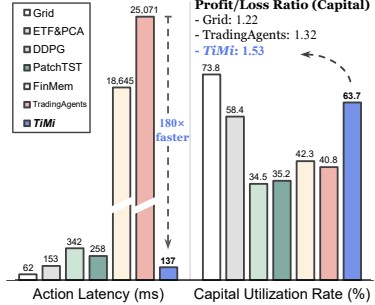

Figure 3: **Comparison of action latency** (left) and **capital utilization** (right) for representative methods.

Figure 4: **Comparative performance (ARR) distributions** of different methods across trading pairs.

of 6.4%, 8.0%, and 13.7% across U.S. stock index futures, mainstream cryptocurrencies, and altcoin markets, respectively. Notably, our system demonstrates stable risk-adjusted returns with promising Sharpe&Sortino Ratios and competitive drawdown control, indicating robust trading sustainability. Crucially, the minute-level trading frequency enables our deployed bots to capitalize on short-term market inefficiencies that daily-frequency methods necessarily overlook. Furthermore, *TiMi*'s extensive market coverage ($N_{\mathcal{P}} = 213$) matches that of quantitative approaches while surpassing previous ML&RL and agent methods, which typically support fewer trading pairs due to convergence challenges and data requirements for trading action (evidenced in Table 2, m/h/d: minute/hour/day). These empirical results thus confirm the transformation of our rationality-driven paradigm towards demonstrable trading efficacy in market dynamics.

**Action efficiency and capital management.** On the left side of Figure 3, we record the inference-only time of one action cycle per trading pair. Benefiting from architectural decoupling, our *TiMi* achieves latency on par with quantitative methods, which is fundamentally unattainable for continuous-model-inference approaches. On the right side, we calculate Capital Utilization Rate as $\text{avg}_{\mathcal{P}}\left(\frac{\text{deployed capital}}{\text{available capital}}\right)$. *TiMi* shows clear advantages among learning-based approaches, indicating the ability to capitalize on a broader range of trading opportunities while maintaining strategic position sizing. Additionally, we provide the ratio between profits/losses generated per unit of invested capital, and *TiMi* possesses a competitive ratio of 1.53, outperforming both Grid (1.22) and TradingAgents (1.32) approaches. This metric is significant as it quantifies the efficacy to balance profitable and loss-making trades.

Table 4: Component-wise ablation studies simulated within the 2024 crytocurrency market.

| Method | Configuration | ARR% ↑ | SR ↑ | MDD% ↓ | Live Deployment |
|---|---|---|---|---|---|
| *TiMi* | *full system* | **20.9** | **1.23** | **15.3** | *stable* |
| $\mathcal{A}_{fr}^{\dagger}$ variant | *parameter-only optimization* | 12.5 | 0.92 | 16.3 | *logic inconsist* |
| $\mathcal{A}_{fr}^{\ddagger}$ variant | *semantic-only reflection* | 1.1 | 0.05 | 25.1 | ***stable*** |
| w/o $\mathcal{A}_{sa}$ | *unified strategy* | 15.2 | 0.95 | 28.4 | ***stable*** |
| w/o $\mathcal{A}_{fr}$ | *prototype bot* $\mathcal{B}$ | 1.1 | 0.05 | 25.1 | *runtime unstable* |
| w/o $\mathcal{A}_{sa}$ & $\mathcal{A}_{fr}$ | *minimal baseline* | -4.5 | -0.21 | 34.2 | *runtime unstable* |

## 4.3 ANALYTICAL STUDY

**In-depth analysis of performance distribution.** According to the distribution results in Figure 4, most significantly, *TiMi* exhibits markedly *performance stability* with reduced variance ($\sigma = 11.03\%$) and rare tail events ($<2\%$), indicating more consistent returns over market dynamics compared to alternatives. This characteristic is particularly valuable in algorithmic trading where catastrophic drawdowns often negate long-term performance advantages. It is evident when contrasting with RL approaches like DDPG, which despite showing competitive median returns, suffers from extreme volatility ($\sigma = 29.64\%$) that undermines its reliability in practical deployment. The rationality-driven multi-agent design of *TiMi* appears to effectively *navigate the inherent trade-off* between return maximization and risk minimization that challenges trading systems, achieving a more favorable risk-adjusted profile through its hierarchical optimization and mathematical reflection.

**Ablation study of agents and optimization.** Employing the macro analysis agent $\mathcal{A}_{ma}$ and bot evolution agent $\mathcal{A}_{be}$ as the operational backbone, we conduct granular component-wise ablation experiments simulated in the 2024 crytocurrency market to isolate the contributions of agent specialization (*i.e.*, $\mathcal{A}_{sa}$ and $\mathcal{A}_{fr}$) and our optimization mechanism. As quantified in Table 4, removing $\mathcal{A}_{sa}$ nearly doubles Maximum Drawdown to 28.4%, underscoring its critical role in harmonizing diverse assets for consistent risk exposure (*e.g.*, stable utility tokens vs. high-volatility meme coins). Regarding optimization, the semantic-only $\mathcal{A}_{fr}^{\ddagger}$ ensures stability via syntax resolution yet stagnates in profitability. Conversely, the parameter-only $\mathcal{A}_{fr}^{\dagger}$ yields theoretical gains but proves *logically inconsistent* for live

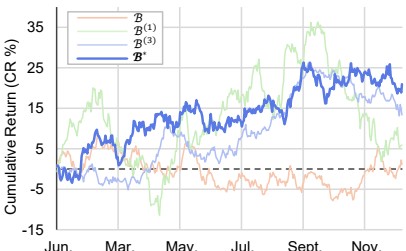

Figure 5: **Comparison of trading bot variants**: $\mathcal{B}$, $\mathcal{B}^*$, and their intermediate versions (1/3-cycle optimization), simulated in 2024 cryptocurrency markets.

deployment due to code-parameter dissonance, while the unoptimized baselines succumb to *runtime instability*. Consequently, only the full system sustains the *policy-deployment-optimization* chain, a synergy corroborated by the evolutionary trajectory in Figure 5; here, the unoptimized $\mathcal{B}$ stagnates near break-even and $\mathcal{B}^{(1)}$ suffers degradation despite transient 35% peaks, exposing the brittleness of shallow parameter tuning without structural adaptation, whereas the stabilized $\mathcal{B}^{(3)}$ culminates in $\mathcal{B}^*$ achieving consistent returns over 20%, validating the iterative efficacy of constraint-based solving and hierarchical interventions detailed in Section 2.5 and Figure 2.

**Transaction visualization.** We present empirical evidence of *TiMi* efficacy upon minute-level transactions across four representative cryptocurrency pairs visualized in Figure 6. The candlestick charts illustrate the adaptive order strategy implemented in $\mathcal{B}^*$, with buy (↑) and sell (↓) indicators precisely demarcating transaction points. Notably, higher-volatility pairs such as SIGN/USDT (82.21%) and OM/USDT (74.39%) yield superior profitability metrics (+32.75% and +10.78% PnL respectively) with correspondingly higher order densities (39 and 28 valid orders), thereby demonstrating the capability of the system to capitalize on price oscillations. Conversely, lower-volatility assets like TRUMP/USDT and XRP/USDT have more conservative trading patterns. These visualizations substantiate that the parameter matrices $\mathbf{M}_P$ and $\mathbf{M}_Q$, tuned by deep optimization cycles with mathematical feedback, can effectively *modulate order execution intensity* to pair-specific volatility while maintaining robust risk management over diverse market conditions, including sustained directional movements, consolidation phases, and extreme price actions.

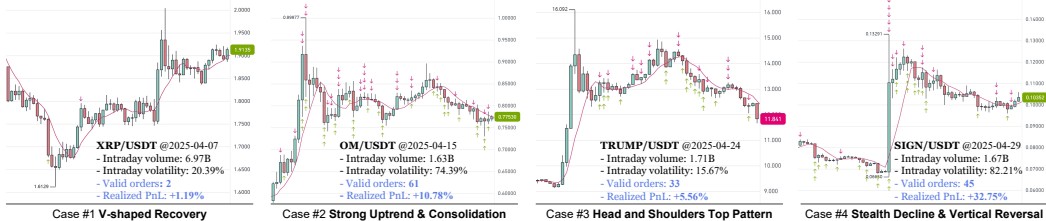

Figure 6: **Detailed transactions of *TiMi* on four representative cryptocurrency trading pairs.** These 15-minute candlestick charts display market movements with green candles indicating price increases and red candles showing price decreases. Buy (↑) and sell (↓) actions performed by *TiMi* are marked on each chart, demonstrating its robust trading capabilities across various market dynamics including uptrends, downtrends, consolidation periods, and extreme price movements.

## 5 RELATED WORK

**LLM-powered agentic system.** Agentic systems built upon LLMs can be categorized into agentic workflows and autonomous agents (Zhuge et al., 2023; Hong et al., 2024a; Zhang et al., 2024b) by autonomy level. The former follows predefined processes with multiple LLM invocations, while the latter employs flexible decision-making. Agentic workflows can be broadly categorized into general and domain-specific types. Workflows further separate into general approaches (Wei et al., 2022; Madaan et al., 2023) and domain-specific ones (*e.g.*, code generation (Hong et al., 2024b; Zhong et al., 2024a), data analysis (Xie et al., 2024; Li et al., 2024a), and mathematical problem-solving (Zhong et al., 2024b; Xin et al., 2024)). Research advances agentic optimization through automated prompt optimization (Fernando et al., 2024; Yang et al., 2024), hyperparameter optimization (Saad-Falcon et al., 2024), and workflow optimization (Hu et al., 2024; Zhang et al., 2025). The proposed *TiMi* for financial trading exemplifies domain-specific implementation, while its hierarchical reflection provides insights into agentic optimization, and we will continuously explore the potential of our rationality-driven agentic system as a generalist.

**Agents for financial trading.** Financial trading agents fall into three architectures (Ding et al., 2024): news-driven, reflection-driven, and factor optimization frameworks. News-driven agents (Zhang et al., 2023; Wang et al., 2024a) incorporate up-to-date news and events to make informed decisions, with approaches like FinMem (Yu et al., 2024a), FinAgent (Zhang et al., 2024c), and CryptoTrade (Li et al., 2024b). Reflection-driven agents (Xing, 2025; Koa et al., 2024) enhance decisions through reflection and debating. For instance, StockAgent (Zhang et al., 2024a) and TradingAgents (Xiao et al., 2025) implement multi-agent frameworks to simulate investor trading behavior and conduct role-based collaboration, and Fincon (Yu et al., 2024b) introduces conceptual verbal reinforcement to refine decision making. Beyond direct trading, other agents (Wang et al., 2024b; 2023) function as alpha factor optimizers for quantitative strategies. In this paper, we harmonize the strategic depth of agents with the mechanical rationality expected for quantitative trading, and pioneer a decoupled paradigm emphasizing progressive strategy development and quantitative-level deployment.

## 6 CONCLUSION

In this paper, we present *TiMi*, a multi-agent system designed with mechanical rationality for algorithmic trading that decouples complex analysis from time-sensitive execution. Through a three-stage process (policy, optimization, and deployment), *TiMi* demonstrates promising and stable performance across diverse financial markets. Our key innovations lie in: (1) a multi-agent architecture leveraging specialized LLM capabilities in semantic analysis, code programming, and mathematical reasoning; (2) a decoupling mechanism separating analysis from deployment; (3) a two-tier analytical paradigm from macro patterns to micro customization; (4) a layered programming design for trading bot implementation; and (5) a closed-loop optimization system driven by mathematical reflection.

**Limitations and ethics statement.** The necessity of the optimization stage limits the zero-shot performance of trading bots developed by *TiMi* when porting to new markets. From a broader perspective, advancements in automatic trading systems may affect market dynamics and liquidity provision, while issues of market fairness and accessibility may also widen the gap between institutional and retail investors. We aim to explore the development of customizable agentic trading systems, and this paper does not constitute investment advice — *investment is risky, be cautious before entering*.

## 7 ACKNOWLEDGMENT

This work was supported by National Natural Science Fund of China (No. U25A20527, 62473286). This work was also supported by Shanghai Municipal Science and Technology Major Project (No. 2025SHZDZX025G10).

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

## A DELVE INTO MATHEMATICAL REASONING FOR PARAMETER SOLVING

In this section, we provide three *practical* cases of *TiMi* implementation for $\mathcal{B}^*$. Each case illustrates a distinct mode commonly encountered in algorithmic trading. Through these representative scenarios, we aim to show how *TiMi* converts qualitative risk into quantitative optimization, thereby bridging the gap between observed trading pathologies and systematic parameter refinement.

### A.1 CASE #1: POSITION SIZE CONTROL UNDER MARKET VOLATILITY

After a simulation period, the system collects feedback $\mathcal{F}$ for a trading bot operating on OM/USDT pair. The feedback (organized in *structured* data) includes: (1) *performance metrics* of final return, max drawdown, and Sharpe Ratio; (2) *trade logs* with detailed records of valid transactions and positions; (3) *market data* including K-lines for OM/USDT (typically including minute-level and hourly data), volatility, liquidity, funding rate changes, and market capitalization.

The feedback records that the bot incurred a significant drawdown over 50% during a sharp, 30-minute market downturn. An excessively dense series of buy orders was executed as the price fell, leading to an oversized and deeply underwater position. The feedback reflection agent $\mathcal{A}_{fr}$ analyzes this feedback and identifies a risk scenario $\mathcal{R} = \gamma(\mathcal{F})$ that *the order density and size do not adapt sufficiently to sudden volatility spikes*.

Subsequently, $\mathcal{A}_{fr}$ translates this risk scenario into a formal mathematical constraint (linear programming problem), with the goal of limiting potential losses in similar future cases. As illustrated in Section 3, the relevant parameters appear to be the order quantity distribution matrix $\mathbf{M}_Q = [q_1, q_2, ..., q_m]$ with the capital allocation $A$, and the quantity for the $i$-th level order is $Q_i = A \times \mathbf{M}_Q[i] \times c_m \times c_f$. Consequently, the agent can establish a direct constraint on the total position size, where the size of all filled buy orders under the extreme scenario must not exceed a maximum size $Q_{max}$. Specially, we obtain a linear inequality for the parameters $q_i$:

$$\sum_{i=1}^{m} Q_i \leq Q_{max} \implies \sum_{i=1}^{m} q_i \leq \frac{Q_{max}}{A \times c_m \times c_f} \tag{4}$$

where $Q_{max}$ can be derived from risk tolerance (*i.e.*, determined by global capital and parallel trading volume). This forms a specific variant of the inequality $\mathbf{A}(\mathcal{R})\Theta \preceq \mathbf{b}(\mathcal{R})$ from Equation 3. In this case, the parameter vector $\Theta$ contains the elements $q_i$ to be optimized, the corresponding row in the constraint matrix $\mathbf{A}(\mathcal{R})$ would be $[1, 1, ..., 1]$, and the corresponding value in the constraint vector $\mathbf{b}(\mathcal{R})$ would be $\frac{Q_{max}}{A \times c_m \times c_f}$.

### A.2 CASE #2: ORDER BOUNDARY CALIBRATION FOR PRICE SURGE EVENTS

Following a simulated trading period with a bot on DOGE/USDT, *TiMi* collects structured feedback $\mathcal{F}$, including: (1) *performance metrics* indicating catastrophic portfolio decline and excessive maximum drawdown; (2) *trade logs* showing that the highest-level sell order was triggered while prices continued to surge, resulting in rapidly accumulating losses; and (3) *market data* containing minute-level K-line information capturing the price surge incident.

To start with, the feedback reflection agent $\mathcal{A}_{fr}$ analyzes this feedback and identifies a critical risk scenario $\mathcal{R} = \gamma(\mathcal{F})$: *the upper boundary of the order group was inadequately calibrated for the volatility observed during the failure event*. Then, $\mathcal{A}_{fr}$ translates this risk scenario into a formal mathematical constraint.

According to Algorithm 1, price levels are determined by the order distribution matrix $\mathbf{M}_P = [p_1, p_2, ..., p_m]$. Thus, the agent can establish a constraint on the highest price exponent relative to the absolute peak price during the surge: $P_{before} \times (1 + \Phi)^{p_m} > P_{peak}$. By taking the logarithm, we get the solvable inequality for the parameter $p_m$:

$$p_m > \frac{\log(P_{peak}/P_{before})}{\log(1 + \Phi)} \tag{5}$$

where $P_{peak}$ and $P_{before}$ are extracted from the market data. This constraint establishes an evidence-based lower bound for $p_m$ within the inequality system $\mathbf{A}(\mathcal{R})\Theta \preceq \mathbf{b}(\mathcal{R})$.

A.3 CASE #3: ADAPTIVE PROFIT-TAKING UNDER TRENDING MARKET CONDITIONS

After a simulation period on NQ index futures, the system collects structured feedback $\mathcal{F}$ from the trading bot, comprising: (1) *performance metrics*, including final return, profit factor, and comparison with buy-and-hold return; (2) *trade logs*, including detailed records indicating systematic premature closure of profitable long positions; (3) *market data* K-lines for NQ futures, trend strength indicators (*e.g.*, ADX), and historical volatility across trending versus range-bound periods.

The feedback indicates that the deployed bot, while consistently making small profits, underperformed during a sustained market rally. Its profit factor was high, but the total return was lower than a simple buy-and-hold strategy. An analysis of the trade logs shows that profitable long positions were closed too early, capturing only a fraction of the actual upward price movement. The feedback reflection agent $\mathcal{A}_{\text{fr}}$ analyzes this feedback and identifies a risk scenario (or, an *opportunity cost*) $\mathcal{R} = \gamma(\mathcal{F})$ that *the profit-taking thresholds are overly conservative and not adapted to strong trend persistence*.

Sequentially, $\mathcal{A}_{\text{fr}}$ translates this opportunity cost scenario into a formal mathematical constraint. As discussed in Section 3, the relevant parameters include the profit/loss threshold matrix $\mathbf{H} = [h_1, h_2, ..., h_k]$, which determines the exit points $P_{\text{entry}} \times (1 \pm \Phi)^{\mathbf{H}[i]}$. Thus, the agent can establish a constraint on the minimum profit-taking level, where the first profit target for any position must be set wide enough to capture at least the average price movement observed during prior trending phases.

Next, we obtain a linear inequality for the parameter $h_1$. The first profit-taking price, $P_1 = P_{\text{entry}} \times (1 + \Phi)^{h_1}$, must satisfy:

$$P_1 - P_{\text{entry}} \geq \Delta P_{\text{trend}} \tag{6}$$

where $\Delta P_{\text{trend}}$ denotes the average profitable movement during a market trend. This leads to the inequality $(1+\Phi)^{h_1} \geq 1 + \frac{\Delta P_{\text{trend}}}{P_{\text{entry}}}$, and it can be simplified by taking logs to $h_1 \geq \log_{1+\Phi}(1 + \frac{\Delta P_{\text{trend}}}{P_{\text{entry}}})$.

# B DETAILS ON IMPLEMENTATION SPECIFICS FOR DEPLOYMENT

**Reproducibility statement.** To facilitate replication, we provide further implementation details of the *TiMi* system, covering order specifics, exchange selection, risk control mechanisms, transaction cost modeling, and action latency control. These technical details demonstrate the practical considerations necessary for deploying the system in live trading environments. Crucially, we will open-source the implementation of *TiMi* product for deployment and release the key corner case list (from both back simulation and live deployment).

**Order types and exchange selection.** *TiMi* employs three order types for specific trading functions: (1) LIMIT orders serve as the exclusive mechanism for opening positions based on volatility-derived formulas; (2) TAKE_PROFIT and STOP orders are dynamically placed and cancelled during position monitoring; and (3) MARKET orders are utilized for risk management purposes, including liquidating low-risk positions and executing global profit/loss events. We select top-tier exchanges with high liquidity, *i.e.*, CME for stock index futures and Binance for cryptocurrencies.

**Transaction costs and slippage modeling.** We model and mitigate two primary costs: order fees and periodic funding rates. *TiMi* employs LIMIT orders for entry to capture favorable maker fees and eliminate entry slippage. Beyond static fees, *TiMi* adapts to funding rates on a per-pair basis. For instance, high funding rates trigger an order reduction (even deactivation) to avoid accumulating positions with prohibitive holding costs. And a pre-trade price deviation check is conducted to prevent unintended actions during extreme volatility and potential slippage. Real records of transaction costs can be found in the Supplementary Material.

**Action latency control.** In Table 5, we present detailed records of latency with variance. The primary source of latency and, more importantly, tail latency, is *external*: network I/O associated with RTT to the exchange. In a stable network environment, the action latency of *TiMi* is robust. To track the external issues, *TiMi* has progressively implemented engineering optimizations (*e.g.*, dynamic cache and thread executor as shown in Figure 2) with a series of safeguards, including timeouts, state checks, and circuit breakers.

Table 5: **Decomposition of action latency** (ms) for the deployed *TiMi*.

| Latency Source | Avg. | Std. Dev. | P99 |
|---|---|---|---|
| (1) Market Retrieval | 85 | 12 | 115 |
| **(2) Internal Logic** | **5** | **<1** | **5** |
| (3) Trade Request | 47 | 8 | 65 |
| Total End-to-End | 137 | 15 | 185 |

Table 6: Ablation study of the strategy adaptation agent $\mathcal{A}_{sa}$ simulated in 2025 altcoin markets. $\sigma_{ARR}$ represents the standard deviation of annualized returns across pairs, indicating cross-pair stability.

| Method | Configuration | ARR%↑ | SR↑ | MDD%↓ | $\sigma_{ARR}$%↓ |
|---|---|---|---|---|---|
| *TiMi* | *full system* | 13.7 | 0.86 | 32.8 | **11.0** |
| w/o $\mathcal{A}_{sa}$ | *unified strategy* | 10.4 | 0.71 | 38.2 | 19.5 |

**Failover mechanisms for error handling.** At the function level, all external API interactions are wrapped in exception-handling logic with rate limit management. At the process level, failures in concurrent tasks are isolated to guarantee service continuity. Besides, *TiMi* integrates a price deviation check and periodically clears orphaned orders, preventing erroneous actions under market anomalies. At the system level, state information is fetched directly from the exchange, enabling a stateless execution logic. This allows the bots to recover at any time without losing trading context.

## C   ABLATION ANALYSIS OF AGENT COMPONENTS

Above all, our *TiMi* system is designed as a *highly synergistic* architecture. The agents are not merely a collection of components, their functionalities are deeply interlinked to perform the *policy-deployment-optimization* chain. To further understand the contribution of each agent, we offer a component-wise ablation study below:

- **Macro analysis agent $\mathcal{A}_{ma}$ and bot evolution agent $\mathcal{A}_{be}$:** $\mathcal{A}_{ma}$ provides the initial strategic hypothesis from market data, and $\mathcal{A}_{be}$ translates abstract strategies into executable code, serving as the essential bridge to deployment. Consequently, these agents form the *indispensable backbone* of our *TiMi*, and the absence of them would render the entire system non-operational, precluding their ablation.

- **Strategy adaptation agent $\mathcal{A}_{sa}$:** In Table 6, we further provide the ablation results of $\mathcal{A}_{sa}$ simulated in the altcoin future market (2025). The key insight is that while $\mathcal{A}_{sa}$ provides improvement in risk-adjusted returns, its most critical contribution (lower variance in returns across pairs) is enhancing robustness and performance consistency in a diverse market.

- **Feedback reflection agent $\mathcal{A}_{fr}$:** The efficacy of $\mathcal{A}_{fr}$ is empirically validated by the ablation study in Table 4 and Figure 5. The prototype bot $\mathcal{B}$, lacking the optimization stage, stagnates near break-even performance. In contrast, the advanced bot $\mathcal{B}^*$, progressively refined by $\mathcal{A}_{fr}$, achieves consistent growth and a final return exceeding 20%. Crucially, $\mathcal{A}_{fr}$ achieves parameter solving by operating within the proposed hierarchical optimization scheme. As illustrated in Section 2.5 and visualized in Figure 2, the necessity of this scheme is confirmed by the unstable transient gains of bots after only shallow parameter-level tuning versus the sustained profitability of those refined at higher functional and strategic layers.

## D   THE USE OF LARGE LANGUAGE MODELS (LLMS)

We use LLMs solely for checking grammar and polishing writing. Importantly, LLMs did not contribute to the conception of the research problem or the development of the core methodology.

