# OpenReview forum: "Trade in Minutes! Rationality-Driven Agentic System for Quantitative Financial Trading"
_ICLR.cc/2026/Conference — ICLR 2026 Poster_

### Official Review · Reviewer_QZtH · 2025-11-01

**Soundness:** 3
**Presentation:** 3
**Contribution:** 3
**Rating:** 6
**Confidence:** 4

**Summary:**

This paper introduces TiMi (Trade in Minutes), a multi-agent system designed for quantitative financial trading. This system includes (1) Policy Stage (Offline), a Macro Analysis Agent identifies market patterns from technical data to form "general strategies," which a Strategy Adaptation Agent then customizes for specific trading pairs. (2) Optimization Stage (Offline): a Bot Evolution Agent translates these strategies into programmatic trading bots. These bots are then simulated, and a Feedback Reflection Agent analyzes their performance, using mathematical reasoning to solve for optimal parameters and refine the bots. (3) Deployment Stage (Live): the final, optimized programmatic bots are deployed. These bots are low-latency and execute trades at a minute-level frequency using technical indicators, without requiring any live LLM inference.

**Strengths:**

The core contribution—decoupling offline, high-cognition strategy development (Policy & Optimization) from online, low-latency execution (Deployment)—is a significant and practical innovation. It directly addresses the primary bottleneck (latency and cost) of applying agentic LLM systems to real-time, high-frequency domains.

The "Optimization Stage" is a major strength. The use of a Feedback Reflection Agent to translate simulation feedback (e.g., high-risk scenarios) into formal mathematical constraints (as shown in Appendix A) is a novel and powerful mechanism for feedback, far superior to simple parameter-tuning.

The authors validate their system comprehensively. The experiments span multiple markets (stocks, crypto), a large number of trading pairs (200+), and include strong baselines from three distinct categories (Quant, ML/RL, and LLM-Agents). The results, particularly the low latency (Fig. 3), stable returns (Fig. 4), and effective ablation study (Fig. 5), compellingly support the architectural design.

**Weaknesses:**

The paper describes the agents' functions at a high, conceptual level but provides few implementation details. To be convincing, the authors can show their work. For example, what prompts are used? What does the "general strategy" output by Ama actually look like? Is it text, JSON, or code?

**Questions:**

Is the strategy space limited to grid-trading archetypes, or can your agentic system invent and implement entirely different strategies (e.g., a momentum-crossover strategy or a statistical arbitrage strategy) from scratch?

What is the total offline computational cost to fully train and optimize one bot for a single trading pair, from the Policy stage to the Deployment stage?

---

> ### Author Response · Authors · 2025-11-23
> **Response to Weakness 1 of Reviewer QZtH**
>
> > **Weakness 1:**
> > Agentic implementation details
>
> **Response (1/2):**
> - Thanks for your insightful suggestions! We appreciate your query regarding implementation specifics. Here, we present a comprehensive clarification concerning the communication protocol, runtime environment, and agentic chain.
> - We employ a **Hybrid Communication Protocol** combining XML-based message envelopes with JSON payloads for data exchange. Each agent communication follows a standardized schema where the XML layer encapsulates metadata (e.g., sender/receiver identities, task types, and context domain) and the JSON payload contains domain-specific content. For instance, when *$\mathcal{A}_{\text{ma}}$* transmits identified market patterns to *$\mathcal{A}_{\text{sa}}$*, the XML envelope designates the communication type as `macro_to_adaptation`, and the JSON payload carries structured strategy specifications including statistical significance metrics, technical indicator configurations, and entry/exit conditions.
> - Our agents operate within a **deterministic environment with system-level capabilities**, including local file operations, verifiable exchange API invocations, and web searches. Crucially, we implement **procedural posterior check**. *TiMi* can programmatically verify generated scripts and mathematical solutions in a controlled sandbox to capture results and tracebacks. Consequently, this enforces a validation loop where computational results and parameter derivations must strictly satisfy defined constraints before deployment.
> - We further supplement implementation details for our agentic chain. *$\mathcal{A}_{\text{ma}}$* initiates the task by combining semantic analysis with code programming to determine market regimes. Beyond internal knowledge, *$\mathcal{A}_{\text{ma}}$* can invoke system-level exchange APIs to fetch high-granularity historical data and generate Python scripts to quantitatively analyze market breadth, sector momentum, and volatility trends, producing a structural `StrategyBlueprint`. Subsequently, *$\mathcal{A}_{\text{sa}}$* executes a *semi-fixed* paradigm: conducting background checks (i.e., web searches for project whitepapers and fundamental events) on specific trading pairs while querying exchange APIs for critical metadata such as market capitalization, liquidity depth, and recent volatility risks. Based on these observations, *$\mathcal{A}_{\text{sa}}$* can dynamically adapt parameters (e.g., indicator-driven scaling and reducing `risk_allocation_amount` for low-liquidity assets) and output a validated `PairConfiguration`. Next, *$\mathcal{A}_{\text{be}}$* acts as a specialized engineer and adheres to programming laws $\mathcal{L}$ to generate modular Python files from the configuration JSONs, invoking local linters to guarantee syntax (w/ API) correctness. Finally, *$\mathcal{A}_{\text{fr}}$* leverages mathematical reasoning to process deployment feedback and then solves for the parameter set $\Theta^*$ (exemplified in **Appendix A**). *TiMi* then performs the procedural posterior check mentioned above, verifying that the solved parameters satisfy the feasible region defined by risk constraints $\mathbf{A}\Theta \preceq \mathbf{b}$ before the cycle completes.

---

> ### Author Response · Authors · 2025-11-23
> **Response to Weakness 1 of Reviewer QZtH**
>
> > **Weakness 1:**
> > Agentic implementation details
>
> **Response (2/2):**
> - We provide the following data flow example outlining the prompt structure, the XML-based execution trace involving system-level calls, and the validated JSON output generated by *$\mathcal{A}_{\text{ma}}$*:
>     - *System prompt snippet:*
>     ```text
>     You are an expert Quantitative Market Analyst operating within a high-frequency trading environment. You have direct access to a secure runtime for data analysis and market data retrieval.
>
>     Your main goal is to analyze current market conditions programmatically to derive a robust trading strategy configuration. You must base your decisions on quantitative data obtained through tool execution with theoretical priors.
>
>     <core_directives>
>     1. Chain of Thought: You must delineate your reasoning process using <thought> tags before any action.
>     2. Risk Aversion: If data indicates ambiguous market regimes, prioritize CAPITAL PRESERVATION in your strategy configuration.
>     </core_directives>
>
>     <tool_calling>
>     You have tools at your disposal to solve the analysis task. Follow these rules regarding tool calls:
>     1. ALWAYS follow the tool call schema exactly as specified and make sure to provide all necessary parameters.
>     2. The conversation may reference tools that are no longer available. NEVER call tools that are not explicitly provided.
>     3. After receiving tool results, carefully reflect on their quality and determine optimal next steps before proceeding. Use your thinking to plan and iterate based on this new information, and then take the best next action.
>     </tool_calling>
>
>     <output_protocol>
>     Upon completion, you must wrap your final response in an XML envelope.
>     - The XML root must be <envelope>.
>     - The payload inside must be a valid JSON object adhering to the 'StrategyBlueprint' schema.
>     </output_protocol>
>     ```
>
>     - *Agent action trace:*
>     ```xml
>     <thought>
>     To identify the prevailing market regime, I need to assess volatility and trend strength across major assets. I will fetch the last 30 days of hourly data for BTC-USDT via the system API, then use Python to calculate the Average Directional Index (ADX) and Average True Range (ATR) to quantify trend intensity.
>     </thought>
>
>     <call>
>     tool_code_interpreter.run_script(
>         script="import pandas as pd; import talib;
>                 df = get_market_data('BTC-USDT', interval='1h', limit=720);
>                 adx = talib.ADX(df['high'], df['low'], df['close'], timeperiod=14);
>                 atr = talib.ATR(df['high'], df['low'], df['close'], timeperiod=14);
>                 print(f'ADX_Mean: {adx.iloc[-1]:.2f}, ATR_Normalized: {(atr.iloc[-1]/df['close'].iloc[-1]):.4f}')"
>     )
>     </call>
>
>     <observation>
>     STDOUT: ADX_Mean: 42.50, ATR_Normalized: 0.0350
>     STDERR: None
>     </observation>
>     ```
>     - *Inter-agent communication:*
>     ```xml
>     <envelope type="macro_to_adaptation" sender="agent_macro_analysis" receiver="agent_strategy_adaptation" context_domain="crypto_futures">
>     {
>         "strategy_meta": {
>         "type": "Trend_Following_Momentum",
>         "regime_classification": "High_Volatility_Expanding",
>         "statistical_significance": {
>             "adx_score": 42.50,
>             "confidence_interval": 0.95
>         }
>         },
>         "technical_configuration": {
>         "time_window": "15m",
>         "indicators": [
>             {"name": "Keltner_Channels", "params": {"period": 20, "multiplier": 2.0}},
>             {"name": "RSI", "params": {"period": 14, "thresholds": [30, 70]}}
>         ]
>         },
>         "entry_exit_conditions": {
>         "description": "Breakout entry with momentum confirmation and volatility-based stop",
>         "entry_trigger": "price > keltner_upper and (rsi > 60 and rsi < 80)",
>         "stop_loss_type": "trailing_atr",
>         "atr_multiplier": 3.0,
>         "take_profit_type": "risk_reward_ratio",
>         "min_rr_ratio": 2.5
>         }
>     }
>     </envelope>
>     ```

---

> ### Author Response · Authors · 2025-11-23
> **Response to Question 1 of Reviewer QZtH**
>
> > **Question 1:**
> > Can the agentic system implements different strategies from scratch?
>
> **Response:**
> - Yes, the proposed system is architecture-agnostic regarding strategy types and is **capable** of inventing and implementing diverse trading strategies from scratch.
> - We emphasize that our system operates without hard-coded logic and instead directs specialized agents to identify market patterns with statistical significance. Crucially, our **layered programming design** (**Section 2.4**) separates the *strategy layer* (logic) from the *function layer* (execution/data). This architectural modularity enables *$\mathcal{A}_{\text{be}}$* to rewrite the decision-making core from scratch to implement paradigms such as statistical arbitrage or momentum crossovers while seamlessly **reusing underlying functional modules** like data fetching and order routines. Furthermore, this flexibility is encoded in the evolution map (**Figure 2**) through optimization cycle C4 which targets the strategy layer. In scenarios where *$\mathcal{A}_{\text{fr}}$* determines that a strategy archetype is fundamentally unsuited to the prevailing market environment, such as a grid strategy failing during a unidirectional market crash, the system is programmed to escalate intervention to the strategy layer and trigger the re-generation of logical structures rather than merely adjusting parameters within the existing framework.
> - We adopt the dynamic grid strategy as a primary case study because its inherent complexity serves as a superior *stress test* for validating the mechanical rationality and risk control capabilities of *TiMi*. Unlike momentum signals, grid trading necessitates sophisticated multi-order management and utilizes precise mathematical constraints for parameter solving. While *TiMi* is capable of generating various strategies, our extensive evaluations identified the dynamic grid strategy as the optimal performer for the targeted volatility profiles. To share a detail from our ongoing live deployment, we are currently operating specialized sub-versions of this evolved strategy, including a *conservative version* designed to withstand large-scale fluctuations and an *aggressive variant* incorporating micro-scalping logic to capture minimal price discrepancies.

---

> ### Author Response · Authors · 2025-11-23
> **Response to Question 2 of Reviewer QZtH**
>
> > **Question 2:**
> > Offline computational cost from policy to deployment
>
> **Response:**
> - Thank you for this valuable question regarding the computational economics of our system. The total offline computational expenditure for constructing a single *TiMi* bot constitutes a one-time fixed investment that is heavily concentrated in the optimization stage, as the inference cost for generating prototypes in the policy stage is negligible. This optimization phase executes a rigorous architectural evolution alongside parameter tuning. To quantify, the system typically evolves from a minimal logical prototype of approximately 100 lines into a production-ready executable exceeding 2,000 lines, incorporating comprehensive error handling and modular design to ensure deployment robustness. Using our hybrid implementation of commercial APIs for reasoning and local backbones for coding, the average token consumption reaches approximately **0.7 million** per converged bot.
> - We recommend extensive reflection through **6 to 12 months** of historical backtesting and a minimum of **7 days** for live simulation, specifically refining over 90 parameters across more than 10 dimensions. Fun fact: Our aggressive testing across hundreds of pairs once overwhelmed the exchange's regulatory protocols, earning us a 12-hour IP ban (:D).

---

### Official Review · Reviewer_V2m2 · 2025-11-02

**Soundness:** 2
**Presentation:** 3
**Contribution:** 2
**Rating:** 4
**Confidence:** 4

**Summary:**

This paper proposes TiMi, a rationality-driven multi-agent trading system that leverages LLM specialization across three dimensions: semantic analysis (ϕ), code programming (ψ), and mathematical reasoning (γ). The core idea is to decouple strategy analysis from deployment, building trading bots offline through multi-agent reasoning (macro → micro → bot evolution) and deploying lightweight, deterministic bots for real-time trading. The architecture includes:

Strategic decoupling of reasoning and execution.

A two-tier analytical paradigm (macro pattern discovery → pair-specific customization).

A layered programming design for bot implementation.

A closed-loop optimization mechanism based on mathematical reflection (Linear Programming-style constraints).

Empirical results across 200+ trading pairs show better profitability and latency compared to rule-based, ML, and prior LLM-agent systems.

**Strengths:**

1） The paper identifies a central challenge in LLM-based trading: continuous multi-agent reasoning causes latency and instability that limit minute-level decision-making. Framing this as a policy–deployment decoupling problem establishes a clear and theoretically relevant research direction.

2）The four-agent design (macro analysis, strategy adaptation, bot evolution, feedback reflection) aligns with the three capability dimensions of semantic analysis, code generation, and mathematical reasoning, yielding a systematic and interpretable framework rather than an ad hoc workflow. It is a relevantly reasonable design.

3) The experiments encompass both stock and cryptocurrency trading markets, which makes the evaluation relatively more comprehensive.

**Weaknesses:**

1) Although the paper focuses on higher-frequency trading (minute-level), the experimental time span of only 3–4 months may be too short to adequately evaluate the system’s performance under diverse market conditions. Extending the time horizon to at least half or a full trading year would provide a more robust assessment.

2) More necessary ablation studies can be needed. The experimental section does not isolate the contribution of the main components. In particular, there is no ablation for LLM specialization (ϕ vs ψ vs γ), for the layered programming design, or for the LP-style reflection; the reported ablation is mainly on optimization depth.

**Questions:**

Same as what I mentioned in the weaknesses.

---

> ### Author Response · Authors · 2025-11-23
> **Response to Weakness 1 of Reviewer V2m2**
>
> > **Weakness 1:**
> > Limited experimental time horizon
>
> **Response (1/2):**
> - Thanks you for these perceptive comments! We sincerely appreciate your constructive suggestion regarding the evaluation horizon. We employ this opportunity to clarify our experimental logic, outline the continued robustness of the deployed *TiMi*, and present comprehensive backtesting data for the full year of 2024.
> - We emphasize that **live trading** serves as the ultimate test of **end-to-end effectiveness** under real liquidity and latency constraints. Unlike daily-frequency strategies, our *TiMi* system operates at minute-level granularity across over 200 trading pairs. Consequently, the 4-month live window generated **millions of decision points and tens of thousands of executed orders** (a real record of *TiMi* transactions can be found in our **Supplementary Material**). Furthermore, the presented period (Jan-Apr 2025) stress-tested the system through diverse market dynamics, including the January *bull run*, February *correction*, and subsequent *volatility*. We discover that *TiMi* maintained consistent Sharpe ratios across these varying conditions, indicating genuine *adaptability* rather than overfitting to specific patterns.
> - Furthermore, *TiMi* remains in *continuous* deployment with a current Cumulative Return of **16.6%** (normalized under cross-margin). We portray a specific case to illustrate our advantage: On October 11th, the cryptocurrency market experienced a severe flash crash where Bitcoin plummeted 10% and numerous altcoins dropped 80-90% within minutes, triggering a mass liquidation event. This collapse preceded the widespread dissemination of attributed news (i.e., potential Trump tariffs). In this high-stress environment, while news-driven agents might struggle with **latency or reflexive dynamics** (e.g., *market foresight*), *TiMi* concluded the day with a net profit of approximately **3%** and a controllable drawdown. This resilience underscores the superiority of our decoupled architecture. By relying on low-latency programmatic bots, *TiMi* can react instantly to market structure without the delay or potential misinterpretation inherent in continuous real-time LLM inference, a crucial advantage in complex, reflexive market scenarios.

---

> ### Author Response · Authors · 2025-11-23
> **Response to Weakness 1 of Reviewer V2m2**
>
> > **Weakness 1:**
> > Limited experimental time horizon
>
> **Response (2/2):**
> - To directly resolve the concern regarding time horizon, we provide a comprehensive backtesting evaluation using the 2024 historical data.
>     | Method | U.S. Stock Index| Mainstream Coin| Altcoin|
>     | :--- | :---: | :---: | :---: |
>     | *Quantitative Methods* | | | |
>     | MACD | $\color{red}{3.8}$ / $\color{blue}{0.42}$ / $\color{teal}{14.5}$ | $\color{red}{12.6}$ / $\color{blue}{0.84}$ / $\color{teal}{18.4}$ | $\color{red}{4.2}$ / $\color{blue}{0.36}$ / $\color{teal}{32.8}$ |
>     | Momentum | $\color{red}{8.0}$ / $\color{blue}{0.73}$ / $\color{teal}{16.8}$ | $\color{red}{15.7}$ / $\color{blue}{0.93}$ / $\color{teal}{21.4}$ | $\color{red}{-2.8}$ / $\color{blue}{-0.40}$ / $\color{teal}{38.3}$ |
>     | Grid Trading | $\color{red}{-2.8}$ / $\color{blue}{-0.35}$ / $\color{teal}{\underline{10.2}}$ | $\color{red}{-8.3}$ / $\color{blue}{-0.72}$ / $\color{teal}{22.6}$ | $\color{red}{9.4}$ / $\color{blue}{0.72}$ / $\color{teal}{18.7}$ |
>     | Pairs Trading | $\color{red}{2.3}$ / $\color{blue}{0.40}$ / $\color{teal}{\mathbf{7.4}}$ | $\color{red}{-6.8}$ / $\color{blue}{-0.67}$ / $\color{teal}{16.3}$ | $\color{red}{-5.7}$ / $\color{blue}{-0.46}$ / $\color{teal}{\mathbf{14.5}}$ |
>     | ETF&PCA | $\color{red}{4.6}$ / $\color{blue}{0.49}$ / $\color{teal}{13.9}$ | $\color{red}{7.2}$ / $\color{blue}{0.62}$ / $\color{teal}{15.8}$ | $\color{red}{6.3}$ / $\color{blue}{0.69}$ / $\color{teal}{\underline{17.2}}$ |
>     | TSMOM | $\color{red}{\mathbf{10.5}}$ / $\color{blue}{\underline{0.81}}$ / $\color{teal}{19.7}$ | $\color{red}{\mathbf{18.4}}$ / $\color{blue}{\underline{1.15}}$ / $\color{teal}{17.2}$ | $\color{red}{1.5}$ / $\color{blue}{0.05}$ / $\color{teal}{41.6}$ |
>     | OFI | $\color{red}{2.1}$ / $\color{blue}{0.34}$ / $\color{teal}{11.8}$ | $\color{red}{8.9}$ / $\color{blue}{0.66}$ / $\color{teal}{19.3}$ | $\color{red}{7.8}$ / $\color{blue}{0.64}$ / $\color{teal}{23.9}$ |
>     | *ML&RL Methods* | | | |
>     | LSTM | $\color{red}{3.2}$ / $\color{blue}{0.36}$ / $\color{teal}{15.3}$ | $\color{red}{9.4}$ / $\color{blue}{0.88}$ / $\color{teal}{17.8}$ | $\color{red}{-16.8}$ / $\color{blue}{-0.89}$ / $\color{teal}{25.4}$ |
>     | DQN | $\color{red}{8.3}$ / $\color{blue}{0.69}$ / $\color{teal}{18.6}$ | $\color{red}{9.7}$ / $\color{blue}{0.83}$ / $\color{teal}{20.5}$ | $\color{red}{-9.3}$ / $\color{blue}{-0.80}$ / $\color{teal}{38.7}$ |
>     | DDPG | $\color{red}{5.4}$ / $\color{blue}{0.52}$ / $\color{teal}{17.4}$ | $\color{red}{14.8}$ / $\color{blue}{1.09}$ / $\color{teal}{\underline{14.0}}$ | $\color{red}{8.6}$ / $\color{blue}{0.65}$ / $\color{teal}{26.2}$ |
>     | Autoformer | $\color{red}{4.9}$ / $\color{blue}{0.43}$ / $\color{teal}{16.5}$ | $\color{red}{13.2}$ / $\color{blue}{0.97}$ / $\color{teal}{16.4}$ | $\color{red}{\underline{17.5}}$ / $\color{blue}{\underline{0.98}}$ / $\color{teal}{24.8}$ |
>     | PatchTST | $\color{red}{6.7}$ / $\color{blue}{0.58}$ / $\color{teal}{18.1}$ | $\color{red}{12.5}$ / $\color{blue}{0.90}$ / $\color{teal}{17.6}$ | $\color{red}{11.2}$ / $\color{blue}{0.82}$ / $\color{teal}{23.5}$ |
>     | *LLM-based Agents* | | | |
>     | FinGPT | $\color{red}{5.8}$ / $\color{blue}{0.57}$ / $\color{teal}{15.7}$ | $\color{red}{6.6}$ / $\color{blue}{0.59}$ / $\color{teal}{19.8}$ | $\color{red}{7.4}$ / $\color{blue}{0.61}$ / $\color{teal}{31.2}$ |
>     | FinMem | $\color{red}{5.2}$ / $\color{blue}{0.54}$ / $\color{teal}{14.8}$ | $\color{red}{11.3}$ / $\color{blue}{0.96}$ / $\color{teal}{17.4}$ | $\color{red}{-8.9}$ / $\color{blue}{-0.60}$ / $\color{teal}{19.6}$ |
>     | TradingAgents | $\color{red}{6.3}$ / $\color{blue}{0.60}$ / $\color{teal}{16.2}$ | $\color{red}{\underline{17.9}}$ / $\color{blue}{1.12}$ / $\color{teal}{20.3}$ | $\color{red}{9.7}$ / $\color{blue}{0.72}$ / $\color{teal}{29.6}$ |
>     | ***TiMi (ours)*** | $\color{red}{\underline{8.9}}$ / $\color{blue}{\mathbf{0.84}}$ / $\color{teal}{10.5}$ | $\color{red}{16.5}$ / $\color{blue}{\mathbf{1.25}}$ / $\color{teal}{\mathbf{12.1}}$ | $\color{red}{\mathbf{23.7}}$ / $\color{blue}{\mathbf{1.27}}$ / $\color{teal}{26.0}$ |
>
>     *Legend: $\color{red}{\text{ARR\%}}\uparrow$ / $\color{blue}{\text{SR}}\uparrow$ / $\color{teal}{\text{MDD\%}}\downarrow$. Bold indicates best; Underline indicates second best.*
> - As presented in the table above (synced to **Table 1** in the revised manuscript), diverse methodologies exhibit distinct advantages. For instance, trend-following methods (e.g., TSMOM) capitalize the strong ETF-driven trends in 2024. Similarly, LLM agents demonstrate competitive performance by leveraging news and potentially *posterior* information available in backtesting. Crucially, *TiMi* achieves a significant *balance* and demonstrates superior *stability* with risk-adjusted returns (SR 1.27 in Altcoins), highlighting its **robustness in managing high-volatility assets** where traditional momentum or pure semantic analysis may struggle.

---

> > ### Comment · Reviewer_V2m2 · 2025-11-27
> >
> > Thanks for the further detailed. That answers my questions. I will increase my score.

---

> > > ### Author Response · Authors · 2025-11-28
> > > **Thank you!**
> > >
> > > Thank you for raising the rating! Your ongoing support is invaluable, and we strongly believe that your insights have strengthened the work.

---

> ### Author Response · Authors · 2025-11-23
> **Response to Weakness 2 of Reviewer V2m2**
>
> > **Weakness 2:**
> > Lack ablation study for LLM specialization, for the layered programming design, or for the LP-style reflection
>
> **Response:**
> - We provide granular ablation studies simulated in the 2024 cryptocurrency markets to delineate the specific contributions of agent specialization, the LP-style reflection mechanism, and the layered programming design.
> - Essentially, the macro analysis agent *$\mathcal{A}_{\text{ma}}$* and bot evolution agent *$\mathcal{A}_{\text{be}}$* form the system's operational backbone, and our ablation focuses on the strategy adaptation (*$\mathcal{A}_{\text{sa}}$*) and feedback reflection (*$\mathcal{A}_{\text{fr}}$*) agents. Simulation results (synced to **Table 4** in the revised manuscript) cover 2024 crypto markets are as follows (*deployment viability* indicates engineering robustness in live environments):
>     | Method | Description | ARR\% $\uparrow$ | MDD\% $\downarrow$ | SR $\uparrow$ | Deployment Viability |
>     | :--- | :--- | :---: | :---: | :---: | :---: |
>     | *TiMi* | *full system* | **20.9** | **15.3** | **1.23** | ***stable*** |
>     | *$\mathcal{A}_{\text{fr}}^{\dagger}$* | *parameter-only optimization* | 12.5 | 16.3 | 0.92 | *logic inconsist* |
>     | *$\mathcal{A}_{\text{fr}}^{\ddagger}$* | *semantic-only reflection* | 1.1 | 25.1 | 0.05 | ***stable*** |
>     | w/o *$\mathcal{A}_{\text{sa}}$* | *unified strategy* | 15.2 | 28.4 | 0.95 | ***stable*** |
>     | w/o *$\mathcal{A}_{\text{fr}}$* | *prototype bot $\mathcal{B}$* | 1.1 | 25.1 | 0.05 | *runtime unstable* |
>     | w/o *$\mathcal{A}_{\text{sa}}$* \& *$\mathcal{A}_{\text{fr}}$* | *minimal baseline* | -4.5 | 34.2 | -0.21 | *runtime unstable* |
>
> - **Analysis of specialization and optimization.** The above results demonstrate that while a generalized strategy remains viable, removing pair-specific customization causes MDD to nearly double (15.3% to 28.4%). This confirms that *$\mathcal{A}_{\text{sa}}$* is critical for distinguishing asset classes (e.g., *stable utility tokens* vs. *high-volatility meme coins*) to ensure consistent risk exposure, which is also evidenced by the 2025 simulation results in **table 6**. Regarding *$\mathcal{A}_{\text{fr}}$*, the semantic-only variant *$\mathcal{A}_{\text{fr}}^{\ddagger}$* secures deployment viability via syntax fixes but stagnates in profitability (1.1% ARR), mimicking the unoptimized prototype. Conversely, parameter-only optimization *$\mathcal{A}_{\text{fr}}^{\dagger}$* improves ARR to 12.5% but fails viability because forcing parameter updates without adapting functional code creates logic conflicts. Only the full chain achieves synergy, evidenced by high-precision parameters like `PROFIT_VOLATILITY` and `DYNAMIC_POSITION_SCALER` that act as digital fingerprints of LP-based optimization. This efficacy is also illustrated in **Figure 5**, where the optimized $\mathcal{B}^*$ attains over 20% CR compared to the stagnant prototype $\mathcal{B}$ and the unstable intermediate $\mathcal{B}^{(1)}$, which degraded after a brief 35% peak.
> - **Evaluation of layered programming design.** We compare our layered programming design (discussed in **Section 2.4** and **Figure 2**) against a monolithic approach. We employ three metrics: *first-pass success rate* (FPSR), measuring the correctness of generated codes that satisfy syntax, logic, and API constraints without revision; *max sustainable cycles* (MSC), quantifying the number of optimization rounds before code manageability collapses; and *verified functional blocks* (VFB), counting the distinct, logically independent components.
>
>     | Paradigm | FPSR\% $\uparrow$ | MSC $\uparrow$ | VFB $\uparrow$ |
>     | :--- | :---: | :---: | :---: |
>     | *Layered* | **87.3** | **$>$ 8** | **38** |
>     | Monolithic| 42.1 | 4 | 15 |
>
> - As shown above, the monolithic approach degrades rapidly (MSC=4) with a low 42.1% FPSR as the LLM coder struggles with context loss. In contrast, our layered design decouples dependencies to sustain deep evolutionary cycles (**$>$ 8 rounds**) and successfully integrates **38 distinct functional blocks**. This modular integrity enables complex engineering capabilities, such as specific `position_monitor` logic, `concurrent.futures` handlers, and granular `RequestException` wrappers, which would otherwise suffer catastrophic regression in a monolithic paradigm.

---

### Official Review · Reviewer_ZzEo · 2025-11-02

**Soundness:** 2
**Presentation:** 3
**Contribution:** 2
**Rating:** 4
**Confidence:** 2

**Summary:**

TiMi is a rationality-driven, multi-agent LLM-based trading system designed for minute-level quantitative trading. TiMi decouples complex strategic analysis (policy and optimization stages) from real-time execution (deployment stage) via a modular framework of specialized agents (semantic analysis, code programming, mathematical reasoning). The system uses macro-to-micro strategy formulation, layered bot programming, and hierarchical optimization.

**Strengths:**

- Clear decoupling between inference and execution phases enables simultaneous low-latency deployment and high-frequency efficiency.
- The closed-loop optimization mechanism based on mathematical reflection (e.g., parameter solving via linear programming) is innovative and holds significant practical value.
- Supports large amounts of assets (NP=213) while demonstrating stable operational efficiency and capital utilization—achievements beyond the reach of most reinforcement learning and large language model approaches.

**Weaknesses:**

- Roles of individual agents (e.g., Afr vs. Asa) lack clear ablation to quantify their individual importance to final performance.
- Unlike some LLM-trading papers that provide human-readable rationales, TiMi’s decisions seem more black-boxed in deployment.

**Questions:**

Can the authors provide ablation studies on agent components (e.g., remove Afr or skip hierarchical optimization) to isolate their impact?

Besides, could TiMi be extended to generate interpretable rationales for its trades?

---

> ### Author Response · Authors · 2025-11-23
> **Response to Weakness 1 & Question 1 of Reviewer ZzEo**
>
> > **Weakness 1 & Question 1:**
> > Lack agent-wise ablation study
>
> **Response:**
> - Thank you for your constructive feedback! We clarify the significance of individual agents by employing granular ablation studies simulated in the 2024 cryptocurrency markets, specifically to delineate the contributions of agent specialization and our reflection mechanism. We consider the macro analysis (*$\mathcal{A}_{\text{ma}}$*) and bot evolution (*$\mathcal{A}_{\text{be}}$*) agents as the operational backbone, thereby focusing our in-depth ablation on the strategy adaptation (*$\mathcal{A}_{\text{sa}}$*) and feedback reflection (*$\mathcal{A}_{\text{fr}}$*) agents. We outline the results in the following table (synced to **Table 4** in the revised manuscript), where *deployment viability* emphasizes engineering robustness in live environments:
>     | Method | Description | ARR\% $\uparrow$ | MDD\% $\downarrow$ | SR $\uparrow$ | Deployment Viability |
>     | :--- | :--- | :---: | :---: | :---: | :---: |
>     | *TiMi* | *full system* | **20.9** | **15.3** | **1.23** | ***stable*** |
>     | *$\mathcal{A}_{\text{fr}}^{\dagger}$* | *parameter-only optimization* | 12.5 | 16.3 | 0.92 | *logic inconsist* |
>     | *$\mathcal{A}_{\text{fr}}^{\ddagger}$* | *semantic-only reflection* | 1.1 | 25.1 | 0.05 | ***stable*** |
>     | w/o *$\mathcal{A}_{\text{sa}}$* | *unified strategy* | 15.2 | 28.4 | 0.95 | ***stable*** |
>     | w/o *$\mathcal{A}_{\text{fr}}$* | *prototype bot $\mathcal{B}$* | 1.1 | 25.1 | 0.05 | *runtime unstable* |
>     | w/o *$\mathcal{A}_{\text{sa}}$* \& *$\mathcal{A}_{\text{fr}}$* | *minimal baseline* | -4.5 | 34.2 | -0.21 | *runtime unstable* |
> - We conclude that while a generalized strategy remains viable, the removal of *$\mathcal{A}_{\text{sa}}$* significantly increases Maximum Drawdown from 15.3% to 28.4%. This underscores the crucial role of *$\mathcal{A}_{\text{sa}}$* in distinguishing asset characteristics, effectively harmonizing stable utility tokens with high-volatility meme coins to maintain *consistent risk exposure* (also evidenced by the 2025 simulation results in **table 6**). Regarding *$\mathcal{A}_{\text{fr}}$*, we discover that the semantic-only variant *$\mathcal{A}_{\text{fr}}^{\ddagger}$* resolves syntax constraints to ensure stability but stagnates in profitability, mimicking an unoptimized prototype. Conversely, parameter-only optimization *$\mathcal{A}_{\text{fr}}^{\dagger}$* improves ARR but fails deployment viability, as forcing parameter updates without adapting functional code creates logic conflicts. Crucially, **our agentic functionalities are deeply interlinked to perform the *policy-deployment-optimization* chain**, and only the full system promotes the necessary synergy with high-precision parameters (e.g., `FLUCTUATION_WEIGHTS` and `FUNDING_RATE_RULES`), which we portray as digital fingerprints of the optimization process. This efficacy is also illustrated in **Figure 5**, where the optimized $\mathcal{B}^*$ attains over 20% CR compared to the stagnant prototype $\mathcal{B}$ and the unstable intermediate $\mathcal{B}^{(1)}$, which degraded after a brief 35% peak.
> - Furthermore, to understand the structural efficacy, we present an evaluation comparing our layered programming design against a monolithic approach using three metrics: *first-pass success rate* (FPSR), measuring the correctness of generated codes that satisfy syntax, logic, and API constraints without revision; *max sustainable cycles* (MSC), quantifying the number of optimization rounds before code manageability collapses; and *verified functional blocks* (VFB), counting the distinct, logically independent components.
>
>     | Paradigm | FPSR\% $\uparrow$ | MSC $\uparrow$ | VFB $\uparrow$ |
>     | :--- | :---: | :---: | :---: |
>     | *Layered* | **87.3** | **$>$ 8** | **38** |
>     | Monolithic| 42.1 | 4 | 15 |
> - We highlight that the monolithic approach degrades rapidly with a low 42.1% FPSR as the LLM struggles with context loss. In contrast, we adopt a layered design that decouples dependencies to sustain deep evolutionary cycles (**$>$ 8 rounds**) and successfully integrates **38 distinct functional blocks**. This modular integrity allows us to employ complex engineering capabilities, involving specific `order_scheduler` logic, `concurrent.futures` handlers, and granular `StatisticsError` wrappers, which we explain would otherwise suffer catastrophic regression in a monolithic paradigm.

---

> ### Author Response · Authors · 2025-11-23
> **Response to Weakness 2 of Reviewer ZzEo**
>
> > **Weakness 2:**
> > Interpretable rationales in deployment
>
> **Response:**
> - We sincerely thank the reviewer for this insightful comment, which touches upon a crucial and deliberate design philosophy of our *TiMi* system. Above all, *TiMi*'s decisions are not black-boxed during deployment. Our deployed bot $\mathcal{B}^*$ provides **continuous stream of technical interpretability** through highly detailed, structured logs capturing the full action state across key dimensions, including `Market` state, `Logic` triggers, `Execution` details, `Monitoring` status and `Risk` assessments. These real-time logs provide an **immediate, auditable trace** of the bot's mechanical rationality, which is the deterministic execution of the deeper strategic rationale established offline.
> - Essentially, while some LLM-powered trading agents prioritize generating human-readable rationales, we have intentionally engineered *TiMi* to **front-load** the complex reasoning into its offline stages. This architectural choice is fundamental to **resolving the critical trade-off between strategic depth and deployment efficiency in quantitative finance**. Specifically, the logic rationale behind *TiMi*'s decisions is **transparently constructed and codified** during the policy and optimization stages:
>     - **Strategy Formulation:** Our macro analysis *$\mathcal{A}_{\text{ma}}$* and strategy adaptation *$\mathcal{A}_{\text{sa}}$* agents explicitly define trading strategies based on technical indicators and pair-specific characteristics.
>     - **Mathematical Reflection:** The feedback reflection agent *$\mathcal{A}_{\text{fr}}$* deconstructs performance data from simulations and formulates concrete mathematical optimization problems (**exemplified interpretably in Appendix A**).
>     - **Programmatic Embodiment:** The bot evolution agent *$\mathcal{A}_{\text{be}}$* translates the refined strategies and mathematically-derived parameters into executable, programmatic trading bots $\mathcal{B}^*$. The evolution map in **Figure 2** visually portrays this hierarchical and auditable refinement process.
> - More significantly, in fact, *TiMi* continues to operate in live trading, and we can highlight the practical significance of our system design during recent extreme market events. On **October 11th**, Bitcoin plummeted over 10% in fifteen minutes, numerous altcoins flash-crashed by 80-90%, and USDe experienced severe de-pegging, triggering one of the largest mass liquidation events in recent cryptocurrency market history. Crucially, this market collapse preceded the widespread dissemination of its attributed cause (i.e., news regarding potential tariffs from Trump), illustrating severe limitations of news-driven, high-latency analysis in the face of **reflexive** dynamics (e.g., *market foresight*). In this high-stress environment, our deployed *TiMi* not only maintained a controllable Maximum Drawdown but also concluded the day with a net profit of approximately **3%** (normalized under a cross-margin account). This resilience underscores the superiority of our decoupled architecture. By relying on pre-validated, low-latency programmatic bots, ***TiMi* can react instantly to market structure without the delay or potential misinterpretation inherent in continuous real-time LLM inference**, a critical advantage in short-timescale, high-intensity market scenarios.
> - To sum up, our paradigm offers dual advantages: First, it **provides mathematical rigor** through constrained optimization, offering **interpretability grounded in quantifiable objectives** rather than subjective linguistic descriptions. Second, it facilitates comprehensive **ex-ante analysis** during strategy development, allowing practitioners to **thoroughly understand and validate the trading logic before deployment, rather than attempting to interpret decisions under the pressure of live market conditions.** We believe this strategic trade-off, validated by its robustness in extreme market conditions, is essential for creating an agentic system that harmonizes the deep reasoning of LLMs with the mechanical rationality and reliability required for effective quantitative trading.

---

> ### Author Response · Authors · 2025-11-23
> **Response to Question 2 of Reviewer ZzEo**
>
> > **Question 2:**
> > Could TiMi be extended to generate interpretable rationales for its trades?
>
> **Response:**
> - Based on real-time logs captured from deployment, *TiMi*'s architecture is inherently suited for generating richer, multi-layered rationales. We envision a dedicated **rationale generation agent** operating periodically (e.g., after a minute-level trading cycle or daily) to perform comprehensive post-mortem analysis without affecting live trading latency. This agent would synthesize two primary information sources:
>     - **Strategic Rationale (*Why*):** Originates from the structured configuration file that our agentic system generates for each deployed bot. This file serves as a complete, machine-readable blueprint of the strategy. Our current production bots are governed by configurations with 90+ parameters spanning 10+ dimensions, including `Environment`, `Scheduler`, `Filter`, `Order Logic`, and `Monitor`. Parsing this file reveals the precise, high level logic, from broad market selection criteria to specific rules of order execution, all quantitatively defined and optimized by our agentic system.
>     - **Execution Rationale (*What*):** Our deployed trading bots produce highly detailed logs capturing the granular, real time context of each action, including market data snapshots, calculated indicator values that triggered the trade, specific reasons for order placement, and precise execution details.
>
> - Next, we can employ the agent to process these inputs for **multi-modal reviews**. It could leverage a semantic LLM to generate a human-readable narrative in a *post-hoc* manner, connecting the high level strategy defined in the configuration file to the specific market conditions documented in the logs. Furthermore, since all market signals are **algorithmically defined and traceable**, the agent can programmatically generate visualizations similar to **Figure 6** in our paper, intuitively overlaying *TiMi*'s transactions onto corresponding market dynamics.

---

> ### Comment · Reviewer_ZzEo · 2025-11-28
>
> Thanks for providing the ablation experiments; they have addressed my questions! I will update my ratings.

---

> ### Author Response · Authors · 2025-11-28
> **Thank you!**
>
> Thank you for your support and valuable feedback! Your insights have been incredibly helpful, and we are excited to incorporate the changes based on your suggestions into our revised paper.

---

### Meta-Review · Area_Chair_C2Vo · 2026-01-07

**Summary:**

The paper introduces a multi-agent LLM based trading system for high frequency trading. Major concerns expressed by reviewers include (1) lack of ablation study of individual agent's role (2) lack of interpretability of the decision (3) experimental time span is too short. The authors addressed all concerns and added the requested ablation tests.

**Reviewer Concerns:**

All concerns are properly addressed. The authors provide all requested ablation study.

**Reviewer Scores:**

Reviewers ZzEo and V2m2 may raise their scores.

---

### Decision · Program_Chairs · 2026-01-26

Accept (Poster)